# Cited4 is a sex-biased mediator of the antidiabetic glitazone response in adipocyte progenitors

Irem Bayindir-Buchhalter[1,†] , Gretchen Wolff[1], Sarah Lerch[1], Tjeerd Sijmonsma[2,‡] , Maximilian Schuster[1] , Jan Gronych[3], Adrian T Billeter[4], Rohollah Babaei[1], Damir Krunic[5], Lars Ketscher[6], Nadine Spielmann[7], Martin Hrabe de Angelis[7,8,9], Jorge L Ruas[6] , Beat P Müller-Stich[4], Mathias Heikenwalder[2], Peter Lichter[3], Stephan Herzig[10,11] & Alexandros Vegiopoulos[1,*]

## Abstract

Most antidiabetic drugs treat disease symptoms rather than adipose tissue dysfunction as a key pathogenic cause in the metabolic syndrome and type 2 diabetes. Pharmacological targeting of adipose tissue through the nuclear receptor PPARg, as exemplified by glitazone treatments, mediates efficacious insulin sensitization. However, a better understanding of the context-specific PPARg responses is required for the development of novel approaches with reduced side effects. Here, we identified the transcriptional cofactor Cited4 as a target and mediator of rosiglitazone in human and murine adipocyte progenitor cells, where it promoted specific sets of the rosiglitazone-dependent transcriptional program. In mice, *Cited4* was required for the proper induction of thermogenic expression by Rosi specifically in subcutaneous fat. This phenotype had high penetrance in females only and was not evident in beta-adrenergically stimulated browning. Intriguingly, this specific defect was associated with reduced capacity for systemic thermogenesis and compromised insulin sensitization upon therapeutic rosiglitazone treatment in female but not male mice. Our findings on *Cited4* function reveal novel unexpected aspects of the pharmacological targeting of PPARg.

**Keywords** adipocyte progenitors; browning; Cited4; glitazones; insulin sensitivity

**Subject Category** Metabolism

## Introduction

The functional status of adipose tissue has emerged as a key determinant for systemic metabolic homeostasis and disease. Obesity and in particular excess visceral fat are prominent risk factors for type 2 diabetes and cardiovascular disease (Cornier *et al*, 2008; Lee *et al*, 2013). The dissociation of obesity and metabolic dysfunction in the paradigms of lipodystrophy and metabolically healthy obesity indicates that it is not the quantity of fat *per se* but the impaired function which underlies the pathogenic process (Vegiopoulos *et al*, 2017). Inadequate lipid metabolism in adipocytes results in increased circulating and ectopically deposited lipids and consequently in lipotoxicity and malfunction of metabolism in multiple organs (Samuel & Shulman, 2012). Chronically, this contributes to the dysregulation of endocrine circuits, insulin resistance, and essentially to the development of type 2 diabetes. However, most of the current treatment options in type 2 diabetes and prediabetes target mainly the symptoms rather than insulin sensitivity and adipose tissue metabolism as causative factors (Soccio *et al*, 2014; Chatterjee *et al*, 2017).

The glitazone drugs of the thiazolidinedione (TZD) class, agonists of the peroxisome proliferator-activated receptor gamma (PPARg), represent an exception in this regard. TZDs act as potent insulin sensitizers, and this action seems to be mediated predominantly by PPARg in adipose tissue (Soccio *et al*, 2014). Although the use of TZDs has strongly declined due to their side effects, PPARg remains a promising target in the prevention and treatment of type 2 diabetes as highlighted by ongoing basic and clinical research

1   DKFZ Junior Group Metabolism and Stem Cell Plasticity, German Cancer Research Center, Heidelberg, Germany
2   Division Chronic Inflammation and Cancer, German Cancer Research Center (DKFZ), Heidelberg, Germany
3   Division of Molecular Genetics, German Cancer Research Center (DKFZ), German Cancer Consortium (DKTK), Heidelberg, Germany
4   Department of General, Visceral, and Transplantation Surgery, University of Heidelberg, Heidelberg, Germany
5   Light Microscopy Facility, German Cancer Research Center (DKFZ), Heidelberg, Germany
6   Department of Physiology and Pharmacology, Molecular and Cellular Exercise Physiology, Karolinska Institutet, Stockholm, Sweden
7   German Mouse Clinic, Institute of Experimental Genetics, Helmholtz Zentrum München, German Research Center for Environmental Health, Neuherberg, Germany
8   Chair of Experimental Genetics, School of Life Science Weihenstephan, Technische Universität München, Freising, Germany
9   German Center for Diabetes Research (DZD), Neuherberg, Germany
10  Helmholtz Center Munich, Institute for Diabetes and Cancer IDC, Neuherberg, Germany
11  Joint Heidelberg-IDC Translational Diabetes Program, Heidelberg University Hospital, Heidelberg, Germany
    †Present address: Wiley-VCH Verlag GmbH & Co. KGaA, Weinheim, Germany
    ‡Present address: LOEWE Center for Cell and Gene Therapy Frankfurt, Department of Medicine, Hematology/Oncology, Goethe University Frankfurt, Frankfurt am Main, Germany
    *Corresponding author. Tel: +49 6221 423585; E-mail: a.vegiopoulos@dkfz.de

(Ahmadian *et al*, 2013; Soccio *et al*, 2014; Banks *et al*, 2015; Chatterjee *et al*, 2017). Thiazolidinediones have pleiotropic effects on adipose tissue, essentially resulting in improved uptake and metabolism of fatty acids and glucose as well as endocrine function (Rangwala & Lazar, 2004; Ye *et al*, 2004; Boden *et al*, 2005; Festuccia *et al*, 2009). Beyond their ability to enhance adipocyte formation and turnover, TZDs promote mitochondrial biogenesis and fatty acid oxidation in human and rodent white adipose tissue and increase its thermogenic potential ("browning"; Okuno *et al*, 1998; Fukui *et al*, 2000; Yamauchi *et al*, 2001; Wilson-Fritch *et al*, 2004; Boden *et al*, 2005; Bogacka *et al*, 2005; Tang *et al*, 2011). Interestingly, the enrichment of pathways of mitochondrial oxidation and lipid metabolism in subcutaneous fat was recently shown to be the most prominent effect of the TZD rosiglitazone on the transcriptome across adipose tissues (Soccio *et al*, 2017). Increased capacity for adipose tissue thermogenesis is generally accepted to be protective against insulin resistance and dyslipidemia but to which extent the regulation of browning by TZDs mediates insulin sensitization remains unclear (Sidossis & Kajimura, 2015).

Thiazolidinediones are potent stimulators of adipocyte progenitor differentiation in human and murine cell culture and promote the formation of beige/brite thermogenic adipocytes (Digby *et al*, 1998; Elabd *et al*, 2009; Petrovic *et al*, 2010; Ohno *et al*, 2012; Ahmadian *et al*, 2013). In the adult organism, adipocyte progenitors mediate the recruitment of new white and beige adipocytes as it occurs upon tissue expansion, cold exposure, or TZD treatment (Tang *et al*, 2011; Hepler *et al*, 2017). Although the core transcriptional network driving adipogenesis downstream of PPARg activation is well established, the factors responsible for depot-, sex-, and stimulus-specific recruitment of progenitors remain to be determined. Moreover, how the context-dependent regulation of progenitors relates to tissue function and insulin sensitization is poorly understood.

In this study, we searched for novel mediators of PPARg activation in defined adipocyte progenitor cells and identified the transcriptional cofactor Cited4 (CREB-binding protein/p300-interacting transactivator with ED-rich tail, [Braganca *et al*, 2002; Yahata *et al*, 2002]). Little is known so far about the physiological function of Cited4, beyond its involvement in the regulation of cardiac hypertrophy (Bostrom *et al*, 2010; Bezzerides *et al*, 2016). We demonstrate that Cited4 promotes the transcriptional program induced by rosiglitazone in differentiating murine and human adipocyte progenitors and that Cited4 deficiency impairs TZD-dependent but not

β-adrenergically stimulated browning specifically in subcutaneous fat. Remarkably, this defect also manifested upon therapeutic rosiglitazone treatment and was associated with reduced insulin sensitization in a sex-specific manner.

## Results

### Cited4 is a target of rosiglitazone in murine and human adipocyte progenitors promoting beige differentiation and Ucp1 expression

We have previously dissected the global transcriptional response of defined immuno-selected Lin(Ter119/CD31/CD45)$^-$Sca1$^+$ adipocyte progenitors to carbaprostacyclin (cPGI$_2$), the stable analogue of prostacyclin and PPARg agonist promoting beige adipocyte differentiation (Vegiopoulos *et al*, 2010; Bayindir *et al*, 2015; Ghandour *et al*, 2016; Babaei *et al*, 2017). To search for novel physiologically relevant factors mediating the effects of PPARg activation in progenitors, we mined time course expression profiles for cPGI$_2$-regulated genes (Bayindir *et al*, 2015). We identified *Cited4* due to a robust but transient induction by cPGI$_2$ during progenitor differentiation (Fig 1A). Treatment with rosiglitazone (Rosi) in place of cPGI$_2$ potentiated the transient induction of *Cited4* (Fig 1B) and this was recapitulated in the mesenchymal progenitor cell line C3H10T1/2 but not in the preadipocyte cell model 3T3-L1 (Fig EV1A and B). In addition, pioglitazone, a TZD currently used as antidiabetic, transiently increased *Cited4* mRNA expression in primary progenitor cells, albeit with overall lower potency compared to Rosi (Fig EV1C). These data demonstrate that *Cited4* is a likely target of TZDs and PPARg in differentiating adipocyte progenitors.

To determine whether the induction of *Cited4* expression is of functional importance for Rosi-stimulated progenitor differentiation and in particular for the oxidative/thermogenic adipocyte phenotype, we examined primary Lin$^-$Sca1$^+$ cells isolated from female *Cited4*$^{-/-}$ knockout mice, lacking the complete *Cited4* coding sequence and Cited4 protein (Appendix Fig S1A and B). Whereas only a trend of reduced mRNA expression of adipogenic marker genes could be detected in *Cited4*$^{-/-}$ cells after 8 days of differentiation, we observed a significant reduction of key thermogenic marker genes, i.e., *Ucp1*, *Cpt1b*, and *Dio2*, with *Ucp1* mRNA decreased by more than threefold (Fig 1C). The effects of *Cited4* knockout on

---

**Figure 1. *Cited4* is a target of rosiglitazone in murine and human adipocyte progenitors promoting beige differentiation.**

A    mRNA expression in FACS-isolated Lin(Ter119/CD31/CD45)$^-$Sca1$^+$ progenitor cells from female mouse subcutaneous fat, differentiated in the presence of 1 μM cPGI$_2$ or vehicle for the indicated time, as determined by expression profiling (*n* = 3, E-MTAB-3693). ****$P$ = 3 × 10$^{-6}$ (Day 2), ****$P$ = 4 × 10$^{-7}$ (Day 4), ****$P$ = 1 × 10$^{-6}$ (Day 6) in 2 × 2 ANOVA with Bonferroni's posttests (cPGI$_2$ vs. vehicle).

B    mRNA expression in MACS-isolated Lin$^-$Sca1$^+$ progenitor cells from female mouse subcutaneous fat, differentiated in the presence of 100 nM Rosi or vehicle for the indicated time, as determined by qRT–PCR (*n* = 4). ****$P$ = 1 × 10$^{-10}$ (Days 1 and 2), **$P$ = 0.001, in 2 × 2 ANOVA with Bonferroni's posttests (Rosi vs. vehicle).

C    mRNA expression in female Lin$^-$Sca1$^+$ cells, differentiated in the presence of 100 nM Rosi or vehicle for 8 days, as determined by qRT–PCR (*n* = 3). *t*-test *Cited4*$^{-/-}$ vs. *Cited4*$^{+/+}$ (Rosi) *$P$ = 0.013 (*Ucp1*), **$P$ = 0.004 (*Cpt1b*), *$P$ = 0.026 (*Dio2*).

D–F  mRNA expression in primary SVF cells from human subcutaneous fat, differentiated in the presence of 100 nM Rosi (D) or vehicle (D–F), as determined by qRT–PCR at the indicated time points (*n* = 5 patients). ♀/♂ represents individual data. (D) ****$P$ = 3 × 10$^{-5}$ (Day 2), ****$P$ = 3 × 10$^{-6}$ (Day 6), ****$P$ = 4 × 10$^{-9}$ (Day 10), ****$P$ = 3 × 10$^{-7}$ (Day 14), in 2 × 2 ANOVA with Bonferroni's posttests (Rosi vs. vehicle). (E, F) Pearson correlation coefficient (*r*) and *P*-value are shown.

G    mRNA expression in primary SVF cells from human female subcutaneous fat transfected with the indicated siRNA prior to differentiation in the presence of 100 nM Rosi for 9 days, as determined by qRT–PCR (*n* = 3). ***$P$ = 0.0002 (*CITED4*), **$P$ = 0.002 (*UCP1*), *$P$ = 0.02 (*UCP1*), *$P$ = 0.035/0.026 (*PPARG*), ***$P$ = 0.0006 (*SLC2A4*), **$P$ = 0.002 (*ADIPOQ*) in one-way ANOVA with Tukey's posttests (vs. siCtrl).

Data information: Data are presented as mean ± SEM except for (D) ♀/♂, (E and F) individual data.

thermogenic markers were comparable in a direct comparison between pioglitazone and Rosi, at least at a higher pioglitazone dose, which was required for the effective stimulation of thermogenic expression (Fig EV1D). Intriguingly, there was no effect of the *Cited4* knockout on the differentiation of progenitors from male

mice despite the induction of *Cited4* by Rosi, indicating a sex-specific requirement (Fig EV1E and F).

We next sought to validate these findings in the human system. Indeed, Rosi treatment during the differentiation of stromal vascular fraction (SVF) cells, freshly isolated from subcutaneous fat, induced

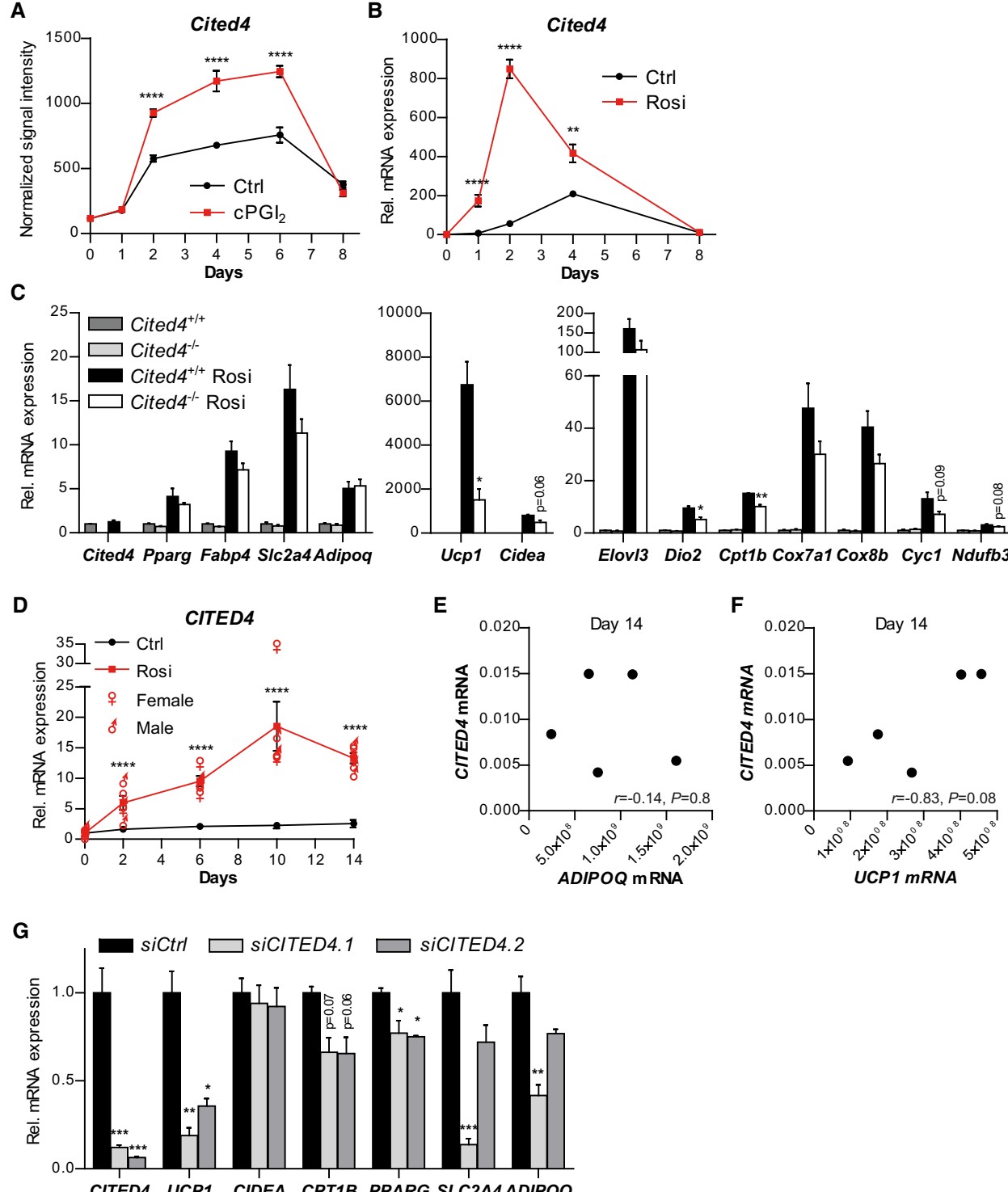

**Figure 1.**

*CITED4* expression by maximally 15-fold independently of donor gender (Figs 1D and EV2A–C). Compared to mouse cells, maximal induction occurred late and declined only marginally at 14 days of differentiation. *CITED4* upregulation paralleled the induction of the general differentiation marker *ADIPOQ* by Rosi (Fig EV2B). However, it is noteworthy that in the absence of Rosi, *CITED4* expression tended to positively correlate with *UCP1* rather than *ADIPOQ* mRNA (Fig 1E and F). Importantly, Rosi-mediated *UCP1* mRNA was markedly diminished in female cells upon *CITED4* knockdown using independent siRNAs and this was accompanied by a milder but significant reduction in *CPT1B* and *PPARG* (Fig 1G), whereas *SLC2A4* and *ADIPOQ* were only affected by one siRNA. In contrast to mouse cells, *CITED4* knockdown resulted in reduced *UCP1* levels in male cells (Fig EV2D). Overall, the Rosi-dependent *CITED4* expression and knockdown phenotype mirrored the murine data and indicate a conserved function of *CITED4* in adipocyte progenitors despite differences possibly attributable to the strong dependency of human adipocyte differentiation on PPARg agonists.

We went on to interrogate the murine phenotype and tested whether it was due to a cell-autonomous function of *Cited4* in progenitors. To this end, we transfected Lin⁻Sca1⁺ cells from *Cited4*$^{F/F}$ mice with Cre recombinase prior to differentiation induction, which resulted in the efficient disruption of the floxed *Cited4* alleles and loss of expression (Fig EV2E and F). In resemblance with the constitutive knockout, Cre-transfected cells showed reduced expression of *Ucp1*, *Cpt1b*, and *Dio2* upon differentiation, with no effect on general adipogenic markers (Fig EV2F). In contrast, transfection of *Cited4*$^{F/F}$ cells with Cre after differentiation induction did not have any considerable effects on differentiation markers (Fig EV2G), suggesting that *Cited4* exerts its essential function in immature progenitors. We further analyzed the consequences of *Cited4* deficiency in immature cells for differentiation by immunofluorescence. Whereas there was no difference between genotypes in the total cell number, we observed a reduction in the number of LipidTOX⁺ adipocytes in *Cited4*$^{F/F}$ + Cre cells by 1/3 (Fig 2A–C). This could be due to the preferential loss of beige adipocytes, given the lack of robust effects on general adipogenic markers. Accordingly, Western blotting revealed a robust reduction in Ucp1 protein in *Cited4*$^{F/F}$ + Cre cells (Fig 2D and E). Finally, we assessed whether *Cited4* knockout could affect mitochondrial respiration as a key function of beige adipocyte metabolism. Basal and maximal mitochondrial respiration was indistinguishable between genotypes arguing against reduced mitochondrial content or a general defect in mitochondrial oxidation (Fig 2F). Treatment with the β3-adrenoreceptor agonist CL-316,243 (CL), which stimulates Ucp1 activity and mitochondrial uncoupling, increased oxygen consumption in wild-type cells. This was abolished in *Cited4*-knockout cells, suggesting a specific defect in uncoupled respiration (Fig 2F and G).

Taken together, these findings argue for an essential function of *Cited4* in the response of adipocyte progenitors to TZD stimulation and in particular in the Rosi-mediated induction of Ucp1.

## Cited4 promotes the transcriptional response to rosiglitazone in adipocyte progenitors

To address the transcriptional pathways regulated by Cited4 in adipocyte progenitors under Rosi stimulation, we performed expression profiling 2 days after differentiation induction at the peak of Rosi-mediated *Cited4* expression (Fig 1B). Ranking of the probe sets/genes by statistical significance resulted in 432 genes with $P < 0.01$ for differential expression between *Cited4*$^{-/-}$ and *Cited4*$^{+/+}$ cells, with 234 down- and 198 upregulated genes in *Cited4*$^{-/-}$ cells. Thermogenic and mitochondrial/oxidative marker genes clustered in the downregulated fraction (Fig 3A). However, besides *Cited4* and *Cyc1*, these genes were either expressed at low levels at day 2 (*Ucp1*, *Cidea*, *Dio2*) or showed $P > 0.01$ for differential expression. Although a substantial proportion of the *Cited4*-dependent genes ($P < 0.01$) were regulated by Rosi ($P < 0.01$ in Rosi vs. Ctrl in wild-type cells), they represented only a minor fraction of the Rosi-regulated transcriptome (Fig 3B). In contrast, our investigation at the level of biological pathways using gene set enrichment analysis (GSEA) showed a substantial overlap between KEGG pathways significantly induced by Rosi and pathways reduced by Cited4 knockout, and vice versa (Fig 3C). In accordance with the phenotype observed at the end of differentiation (Figs 1 and 2), several mitochondrial and lipid metabolism pathways were enriched in the gene fraction downregulated by Cited4 knockout (Fig 3D and E; Appendix Fig S2). In particular, the concerted reduction of genes for oxidative phosphorylation in Rosi-treated *Cited4*$^{-/-}$ cells suggests that Cited4 may promote the induction of oxidative/thermogenic genes at the onset of the differentiation process. In addition, the "PPAR signaling" gene set was among the *Cited4*-dependent gene sets, and furthermore, PPARG was the 2nd highest ranking downregulated gene set in the GSEA for transcription factor motif enrichment (Fig 3F and G; Appendix Table S1). Overall, Rosi treatment doubled the number of pathways significantly affected by Cited4 knockout (Fig 3H). Taken together, the strong interaction between the *Cited4*- and Rosi/PPARg-dependent transcriptional programs suggests that *Cited4* has a specific role in immature progenitors to prime parts of the transcriptional response to Rosi treatment, particularly genes related to beige adipocyte function.

## Cited4 is an essential regulator of thermogenic expression specific to rosiglitazone and subcutaneous fat

To determine the involvement of Cited4 in the Rosi response *in vivo*, we investigated the phenotype of *Cited4*$^{-/-}$ mice. The highest *Cited4* mRNA levels were detected in the heart of both female and male mice (Fig EV3A). In female mice, subcutaneous fat (scWAT) showed comparable levels to skeletal muscle and the highest expression among the fat depots examined including interscapular (BAT) and gonadal (gWAT) fat. Interestingly, *Cited4* expression in scWAT was markedly elevated in female versus male mice, whereas no significant sex-specific differences were observed in other tissues. Cited4 protein was detectable in the heart of wild type but not *Cited4*$^{-/-}$ mice (Appendix Fig S1C). Protein expression was faintly detected in female wild-type scWAT but was clearly observed in scWAT-derived Lin⁻Sca1⁺ progenitor cells (Appendix Fig S1B and D). Examination of *Cited4*$^{-/-}$ mice revealed no difference in body weight and body fat compared to wild type at the onset of adulthood (Appendix Fig S3A). Despite the relatively high expression levels, no physiologically relevant phenotypic effects were observed on heart weight, pulse, or blood pressure (Appendix Table S2).

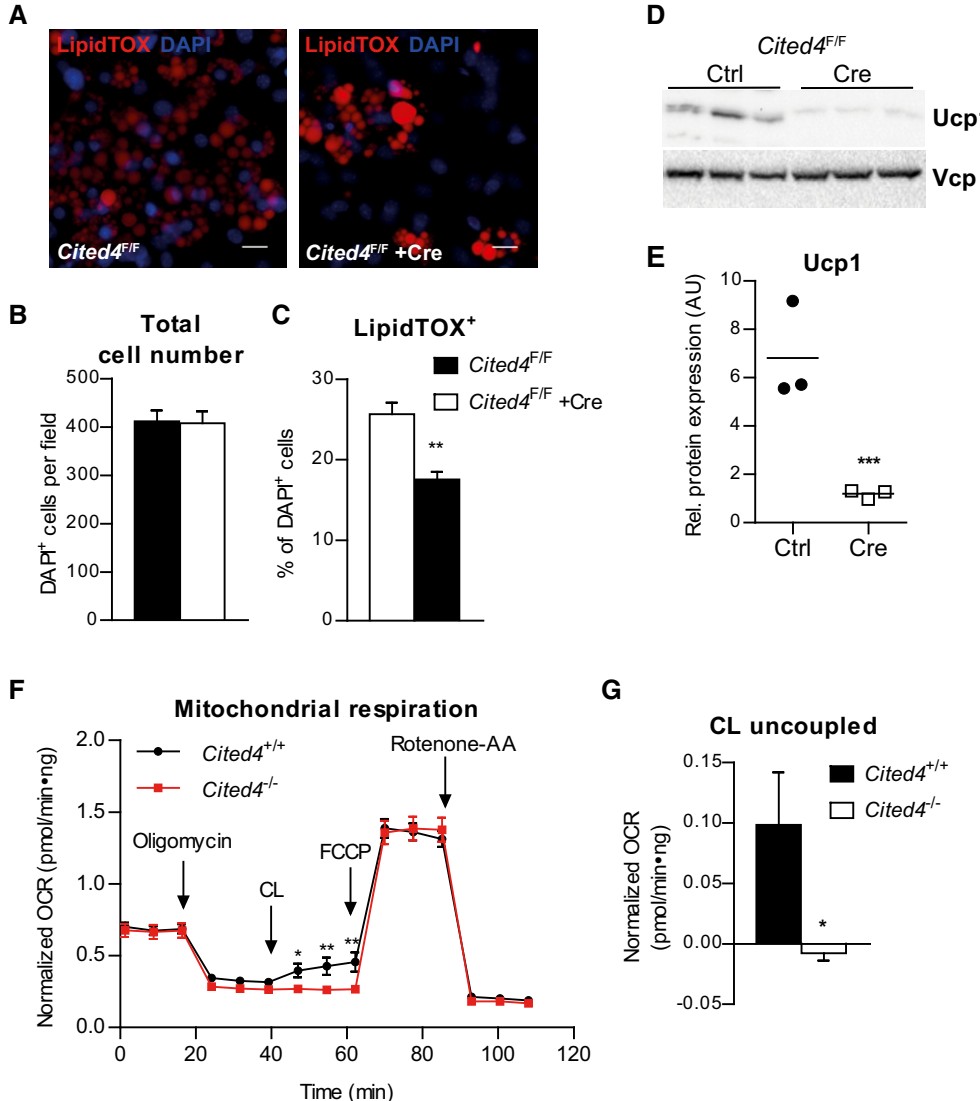

**Figure 2. Cited4 deficiency in progenitors affects Ucp1 protein expression and β3-adrenoreceptor mediated uncoupled respiration.**

A–C  Quantitative fluorescence microscopy of LipidTOX- and DAPI-stained female *Cited4*^F/F Lin⁻Sca1⁺ progenitor cells transfected with Cre or control mRNA prior to differentiation in the presence of 100 nM Rosi for 8 days (*n* = 5). **P = 0.002 in *t*-test (Cre vs. Ctrl). Scale bar is 10 μm.

D, E  Ucp1 expression in female *Cited4*^F/F Lin⁻Sca1⁺ progenitor cells transfected with Cre or control mRNA prior to differentiation in the presence of 100 nM Rosi for 8 days, as determined by Western blot with VCP as loading control (*n* = 3). ***P = 0.0008 in *t*-test (Cre vs. Ctrl).

F  Cellular respiration in female Lin⁻Sca1⁺ progenitor cells differentiated in the presence of 100 nM Rosi for 8 days. The extracellular oxygen consumption rate (OCR) was determined upon injection of the indicated substances and normalized to DNA content. CL: CL-316243. Representative experiment (*n* = 9). *P = 0.041 (47 min), **P = 0.009 (55 min), **P = 0.003 (62 min) in 2 × 2 ANOVA with Holm–Sidak posttests (*Cited4*⁻/⁻ vs. *Cited4*⁺/⁺).

G  Normalized CL-stimulated uncoupled respiration of cells in (F). Values represent the means of the three time points after CL injection after subtraction of the means of the oligomycin time points (*n* = 9). *P = 0.027 in *t*-test (*Cited4*⁻/⁻ vs. *Cited4*⁺/⁺).

Data information: Data are presented as mean ± SEM.
Source data are available online for this figure.

Upon 2.5 weeks of Rosi treatment, female mice did not show differential fat accumulation between genotypes, whereas male *Cited4*⁻/⁻ mice showed mildly increased weight in some fat depots but generally independent of Rosi (Appendix Fig S3B and C). Importantly, there were no differences in the size or number of adipocytes between genotypes, indicating that general adipogenesis and adipocyte turnover were not affected by *Cited4* inactivation (Appendix Fig S3D–G). *Cited4* expression in scWAT was not influenced by

2.5 weeks of Rosi treatment, which could be related to progenitor-selective or an early transient induction (Fig 4A). Rosi induced thermogenic gene expression in subcutaneous fat of wild-type mice but intriguingly, female *Cited4*⁻/⁻ mice showed a 2.5-fold reduction in *Ucp1* expression (Fig 4A), which resulted in markedly compromised Ucp1 protein expression (Fig 4C and D). Further thermogenic markers were affected, including components of the oxidative phosphorylation system (*Cyc1*, *Cox7a1*, *Ndufb3*), mirroring the *ex vivo*

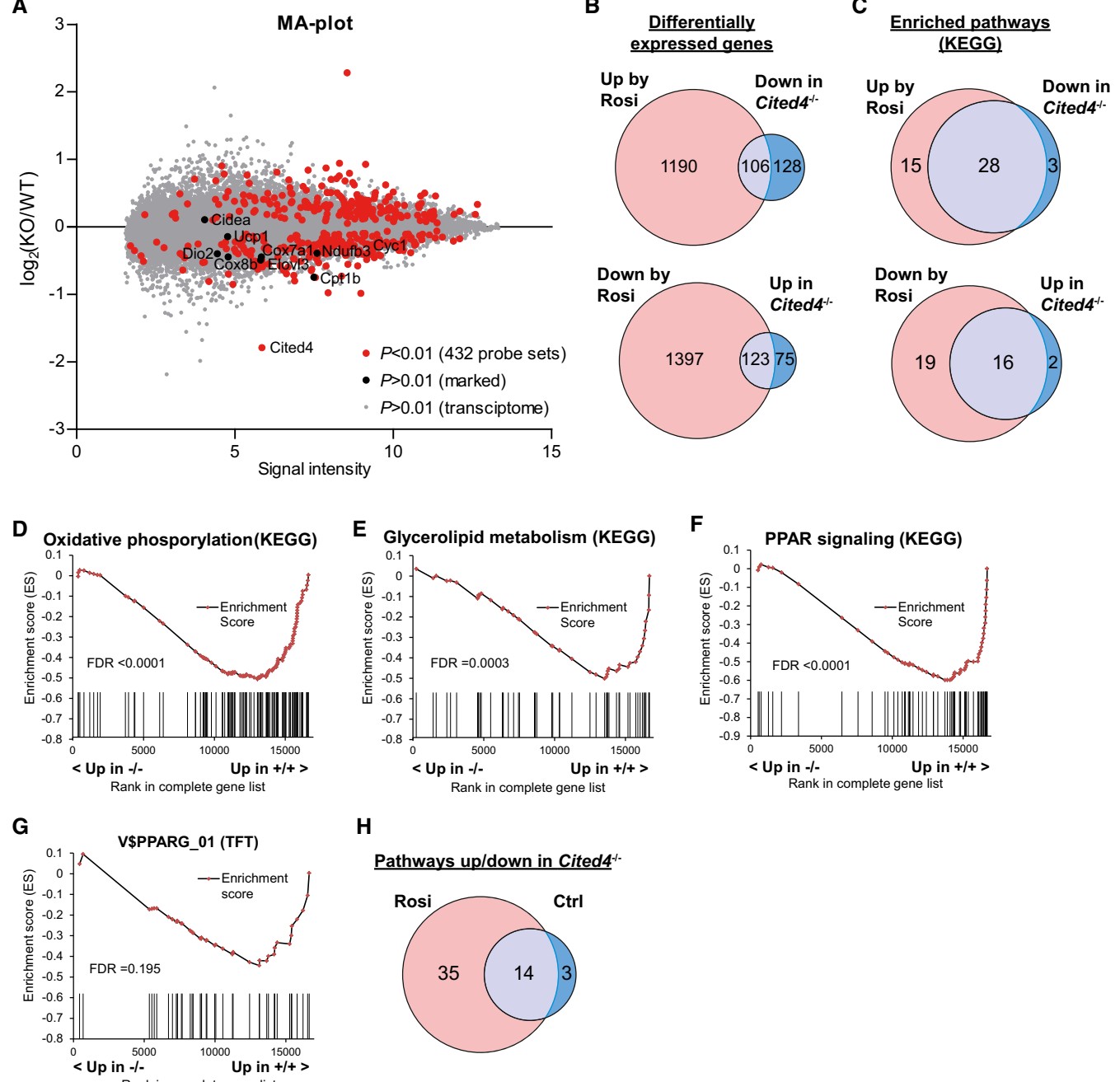

**Figure 3. Cited4- and Rosi/PPARg-dependent transcriptional programs overlap in differentiating adipocyte progenitors.**

A    MA-plot of RNA expression profiles from female Lin−Sca1+ progenitors 2 days after induction of differentiation with 100 nM Rosi, displaying the $\log_2$-ratio of $Cited4^{-/-}$ to $Cited4^{+/+}$ intensities against the average $\log_2$-intensities for all microarray probe sets ($n = 3$). t-test with Welch's correction on $Cited4^{-/-}$ vs. $Cited4^{+/+}$ (Rosi).

B    Comparison of the lists of genes significantly changed by Rosi in wild-type cells ($Cited4^{+/+}$: Rosi vs. Ctrl, $P < 0.01$) or by Cited4-knockout under Rosi treatment (Rosi: $Cited4^{-/-}$ vs. $Cited4^{+/+}$, $P < 0.01$) in expression profiles from female Lin−Sca1+ progenitors 2 days after induction of differentiation with 100 nM Rosi or vehicle ($n = 3$, t-test with Welch's correction).

C    Comparison of the lists of gene sets significantly enriched by Rosi in wild-type cells ($Cited4^{+/+}$: Rosi vs. Ctrl, false discovery rate (FDR) <0.1) or by Cited4-knockout under Rosi treatment (Rosi: $Cited4^{-/-}$ vs. $Cited4^{+/+}$, FDR < 0.1) in GSEA (KEGG) ($n = 3$).

D–G    Enrichment plots from GSEA with the KEGG (D–F) or TFT (G) gene set collection ($Cited4^{-/-}$ vs. $Cited4^{+/+}$), performed on RNA expression profiles 2 days after induction of differentiation with 100 nM Rosi ($n = 3$). Vertical bars represent the individual genes of the gene set.

H    Comparison of the lists of gene sets affected by Cited4-knockout under Rosi or under vehicle treatment ($Cited4^{-/-}$ vs. $Cited4^{+/+}$, FDR < 0.1) in GSEA as in (C).

Data information: Data points in (A) and vertical bars (D–G) represent means for individual genes.

progenitor phenotype. In male knockout mice, thermogenic gene expression was reduced but overall to a lesser extent compared to females (Fig 4B). Although *Ucp1* mRNA was significantly lower in *Cited4*$^{-/-}$ males, Ucp1 protein only showed a trend of reduction, which could be due to the weaker correlation between *Ucp1* mRNA and protein in male compared to female mice (Figs 4C and D, and EV3B; Appendix Fig S4A). Examination of the interscapular (BAT) and gonadal (gWAT) fat depots did not yield any consistent effect of *Cited4* inactivation on thermogenic gene expression (Fig EV3C and D). The question emerged whether *Cited4* is generally essential for thermogenic expression and browning in scWAT. Neither a 10-day treatment with the β3-adrenoreceptor agonist CL-316,243 (CL) nor a 2-week exposure to 5°C resulted in any differences in thermogenic marker expression in scWAT (Fig 4E and F). In conclusion, the data suggest that Cited4 has an essential function in promoting thermogenic expression in scWAT specifically in response to TZD-mediated PPARg activation and in a sex-biased manner.

We next asked whether the compromised Rosi-dependent thermogenic expression in scWAT of *Cited4*-knockout mice could have consequences for systemic energy expenditure. Indirect calorimetry after 2.5 weeks of Rosi treatment revealed a moderate but significant reduction in the oxygen consumption of *Cited4*$^{-/-}$ females in the light phase compared to wild types in the absence of differences in physical activity or food intake (Figs 5A and B, and EV3E and F). This effect was not observed in females on control diet (Fig EV3G). Importantly, a significant reduction could be detected in maximal oxygen consumption upon a single CL injection of Rosi-treated mice, which specifically stimulates brown/beige adipocyte respiration and thermogenesis (Fig 5C and D). In contrast, *Cited4*$^{-/-}$ males showed indistinguishable oxygen consumption under Rosi treatment, both in steady state and upon CL injection (Fig 5E–G). Taken together, the *Cited4*-dependent regulation of thermogenic expression in scWAT is likely to be relevant for the systemic capacity for adipose tissue thermogenesis and therefore systemic metabolism in female but not male mice.

### Cited4 promotes thermogenic expression and insulin sensitization during therapeutic rosiglitazone treatment in females

Adipose tissue and scWAT are considered to be key targets of TZDs for insulin sensitization (Soccio *et al*, 2014, 2017). We sought to determine whether Cited4 is involved in therapeutic Rosi-mediated insulin sensitization and how this associates with the regulation of thermogenic capacity by Cited4 in scWAT. To this end, we fed mice with high-fat diet (HFD) for 16–20 weeks including Rosi treatment in the last 5 weeks. HFD resulted in a 10–15 g gain of body fat mass by week 9 but there were no significant differences in fat accumulation between genotypes either before or after Rosi treatment (Appendix Fig S5A–C) and this was also reflected at the organ level (Appendix Fig S5D and E). Consistent with the phenotype of Rosi-treated mice on normal diet (Fig 4A), key thermogenic and mitochondrial marker genes were expressed at lower levels in scWAT of female *Cited4*-knockout mice, whereas the effects were not significant in male mice (Fig 6A and B). Ucp1 protein expression showed a trend of reduction in *Cited4* knockout females but not in males (Fig 6C and D; Appendix Fig S5F). In addition, the expression of *Cd36* and *Gyk* was affected by *Cited4* inactivation in females,

representing TZD target genes with key functions in lipid metabolism and thermogenic adipocyte function (Rangwala & Lazar, 2004).

Glucose tolerance was mildly but significantly reduced in HFD-fed *Cited4*$^{-/-}$ females upon Rosi treatment but without a concomitant difference in serum insulin in response to glucose, excluding a gross pancreatic defect (Figs 7A and EV4A). In contrast, the beneficial effects of Rosi on fasting insulin were affected by *Cited4* inactivation. Rosi treatment strongly reduced fasting insulin levels in wild type but not in *Cited4*$^{-/-}$ females (Fig 7B). Post-Rosi glucose tolerance and the insulin-lowering effects of Rosi were intact in *Cited4*$^{-/-}$ males (Figs 7C and D, and EV4B). To test whether the observed phenotypic differences in glucose tolerance and fasting insulin could be related to compromised insulin sensitization by Rosi in *Cited4* knockout females, we performed insulin tolerance tests (ITT). Whereas insulin tolerance was not different between genotypes before Rosi treatment, *Cited4*$^{-/-}$ mice had higher blood glucose levels in the rebound phase of the test (Fig 7E and F), which could be confirmed in an ITT with a lower insulin dose (Fig EV4C). Furthermore, the improvement in the overall response of wild-type mice to insulin through Rosi treatment (AUC, $P = 0.055$) was not observed in *Cited4*$^{-/-}$ mice (AUC, $P = 0.568$; Fig 7G). Consistently, there were no genotype effects detected in male mice undergoing the same tests (Figs 7H–J and EV4D). Finally, we observed that the reduction of circulating non-esterified fatty acids (NEFA) in wild-type females by Rosi did not occur in *Cited4*$^{-/-}$ mice (Fig 7K and L), providing a possible link between lipid metabolism and insulin sensitization. In conclusion, *Cited4* deficiency revealed an intriguing sex-specific association between the enhancement of thermogenic capacity and insulin sensitization by therapeutic Rosi treatment.

## Discussion

Targeting adipose tissue and in particular specific functions of PPARg is a promising approach for the treatment of insulin resistance and type 2 diabetes but requires better understanding of the *in vivo* regulation of PPARg responses. Here, we report Cited4 as a context-specific mediator of the TZD response. *Cited4* was required for the proper induction of thermogenic expression specifically in scWAT of female mice, an effect which could be attributed to the regulation of the Rosi-mediated transcriptional program in adipocyte progenitors. Intriguingly, this was accompanied by reduced capacity for thermogenesis *in vivo* as well as compromised therapeutic insulin sensitization. The transcriptional network around PPARg controlling adipogenesis and thermogenic adipocyte function has been characterized to great detail (Lefterova *et al*, 2014; Inagaki *et al*, 2016). However, the molecular factors linking these core processes to the sex-, depot-, and stimulus-dependent differences in adipose tissue composition and metabolism are only beginning to be elucidated.

Extensive evidence suggests that adipose tissue is probably the major target of TZDs for insulin sensitization [reviewed in Ahmadian *et al* (2013), Soccio *et al* (2014)]. Given that compromised insulin sensitivity in *Cited4*$^{-/-}$ mice on HFD was only observed after Rosi treatment, it appears plausible that the insulin-sensitizing function of *Cited4* occurs in adipose tissue. However, *Cited4* was expressed at comparable levels in skeletal muscle, which is a target of TZDs responding with improved insulin sensitivity and

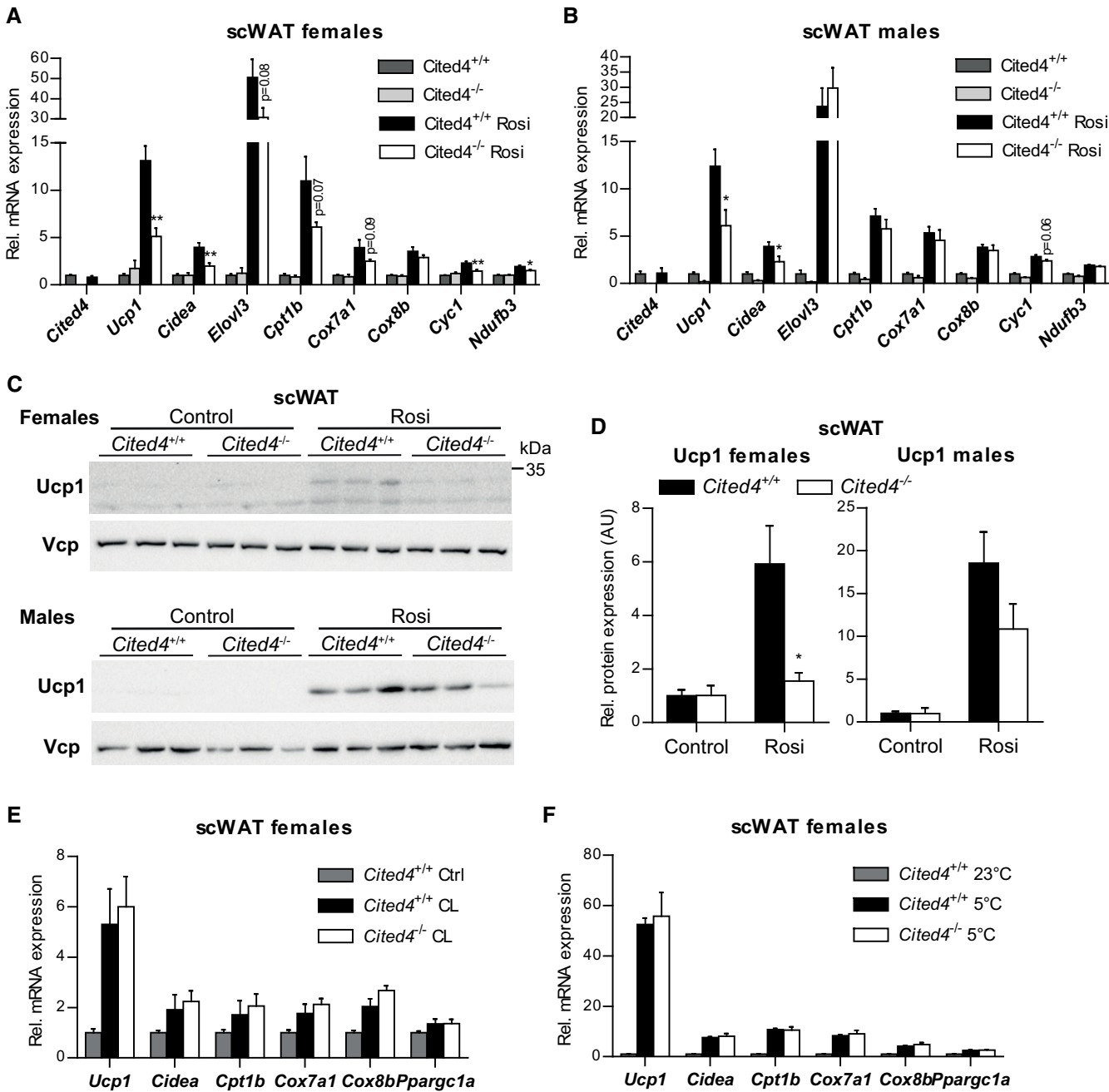

**Figure 4.** *Cited4* **deficiency specifically affects rosiglitazone-mediated thermogenic expression in subcutaneous fat.**

A, B mRNA expression in scWAT of mice fed a diet with 0.0075% Rosi or control diet for 2.5 weeks, determined by qRT–PCR. *t*-test *Cited4*$^{-/-}$ vs. *Cited4*$^{+/+}$ (Rosi), (A) $n = 5/5/6/6$, **$P = 0.003$ (*Ucp1*), **$P = 0.008$ (*Cidea*), **$P = 0.007$ (*Cyc1*), *$P = 0.048$ (*Ndufb3*), (B) $n = 5/4/5/5$, *$P = 0.022$ (*Ucp1*), *$P = 0.037$ (*Cidea*).

C, D Ucp1 expression in scWAT of mice treated as in (A), as determined by Western blot with VCP as loading control. *t*-test *Cited4*$^{-/-}$ vs. *Cited4*$^{+/+}$ (Rosi), females: $n = 5/5/6/6$, *$P = 0.011$, males: $n = 5/4/5/5$.

E mRNA expression in scWAT of female mice treated with CL-316,243 (CL) (1 mg/kg/day via Alzet minipumps) or vehicle for 10 days ($n = 5/7/13$, *t*-test *Cited4*$^{-/-}$ vs. *Cited4*$^{+/+}$ (CL)).

F mRNA expression in scWAT of female mice exposed to 5°C or 23°C for 2 weeks ($n = 5/10/6$, *t*-test *Cited4*$^{-/-}$ vs. *Cited4*$^{+/+}$ (5°C)).

Data information: Data are presented as mean ± SEM.
Source data are available online for this figure.

mitochondrial oxidative potential (Kim *et al*, 2003; Schrauwen *et al*, 2006; Mensink *et al*, 2007; Liu *et al*, 2009). Although the conclusions from two mouse models with muscle-specific *Pparg*

inactivation are contradictory, skeletal muscle Cited4 could contribute to systemic insulin sensitization by Rosi (Hevener *et al*, 2003; Norris *et al*, 2003). Thus, it is not possible to exclude relevant

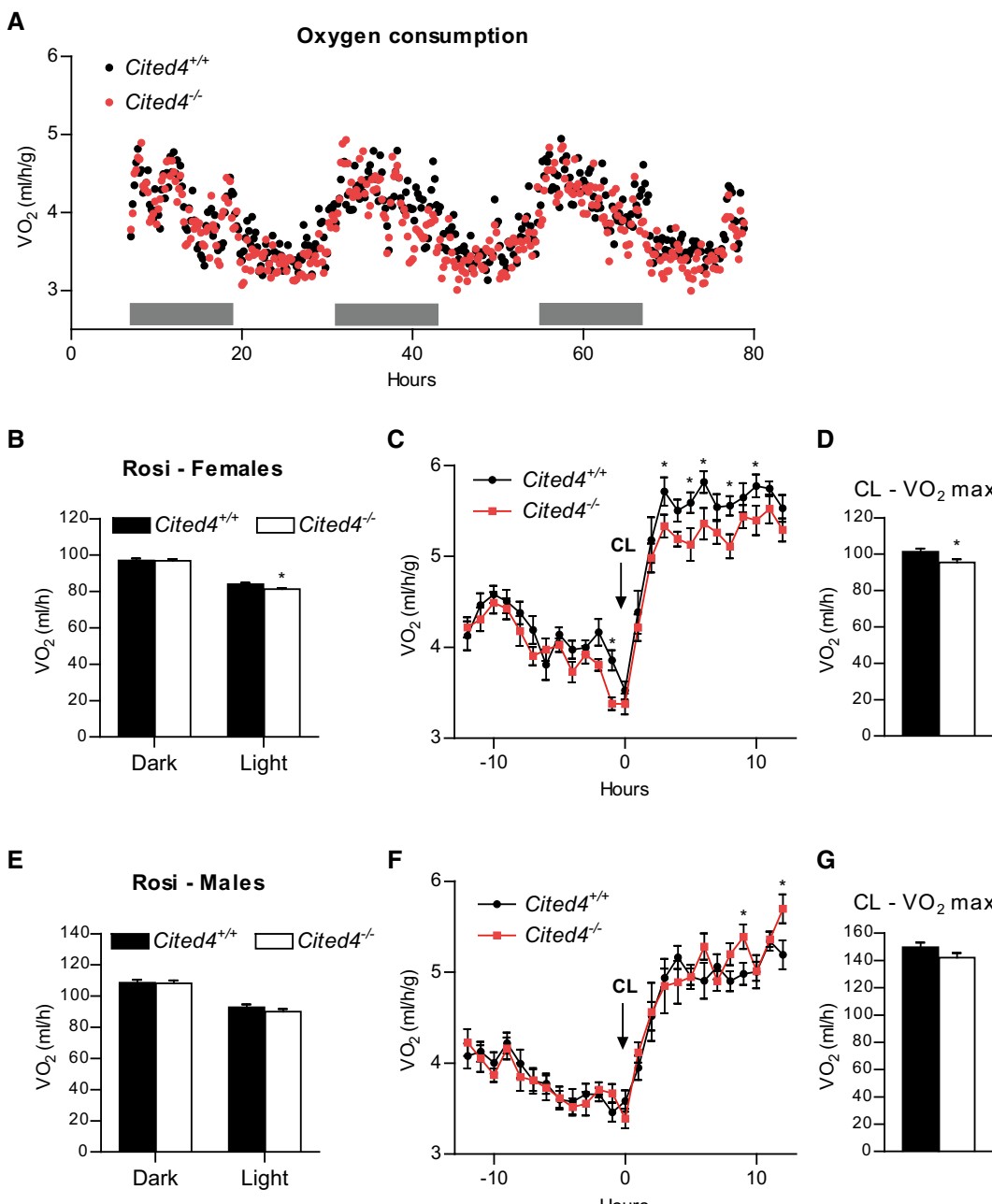

**Figure 5. *Cited4* deficiency affects energy expenditure under rosiglitazone treatment in female mice.**

A  Oxygen consumption rate of female mice fed a diet with 0.0075% Rosi for 2.5 weeks, determined by indirect calorimetry and normalized to body weight (*n* = 9/10). Gray bars represent the dark phase.

B  Oxygen consumption rate of female mice shown in (A) adjusted for body weight by ANCOVA. Three-day averages of $VO_2$ were calculated for each mouse (*n* = 9/10). *$P$ = 0.025 in ANCOVA with Bonferroni's posttest (*Cited4*$^{-/-}$ vs. *Cited4*$^{+/+}$).

C  Oxygen consumption rate of female mice fed a diet with 0.0075% Rosi for 3.5 weeks and injected with 1 mg/kg CL-316,243 (CL) at the indicated time point, as determined by indirect calorimetry (*n* = 9/10). *$P$ = 0.013 (−1 h), *$P$ = 0.045 (3 h), *$P$ = 0.015 (5 h), *$P$ = 0.016 (6 h), *$P$ = 0.019 (8 h), *$P$ = 0.046 (10 h) in repeated-measures 2 × 2 ANOVA with Holm–Sidak posttests (*Cited4*$^{-/-}$ vs. *Cited4*$^{+/+}$).

D  Maximal CL-induced oxygen consumption rate of female mice shown in (C) adjusted for body weight by ANCOVA. Averages of $VO_2$ throughout 3–8 h post-CL injection were calculated for each mouse (*n* = 9/10). *$P$ = 0.049 in ANCOVA with Bonferroni's posttest (*Cited4*$^{-/-}$ vs. *Cited4*$^{+/+}$).

E  Oxygen consumption rate of male mice treated as in (A), determined as in (C) (*n* = 9/10). ANCOVA with Bonferroni's posttest (*Cited4*$^{-/-}$ vs. *Cited4*$^{+/+}$).

F  Oxygen consumption rate of male mice treated as in (C), determined as in (C) (*n* = 9/10). *$P$ = 0.044 (9 h), *$P$ = 0.013 (12 h) in repeated-measures 2 × 2 ANOVA with Holm–Sidak posttests (*Cited4*$^{-/-}$ vs. *Cited4*$^{+/+}$).

G  Maximal CL-induced oxygen consumption rate of male mice shown in (F) and calculated as in (D) (*n* = 9/10). ANCOVA with Bonferroni's posttest (*Cited4*$^{-/-}$ vs. *Cited4*$^{+/+}$).

Data information: Data presented as mean ± SEM.

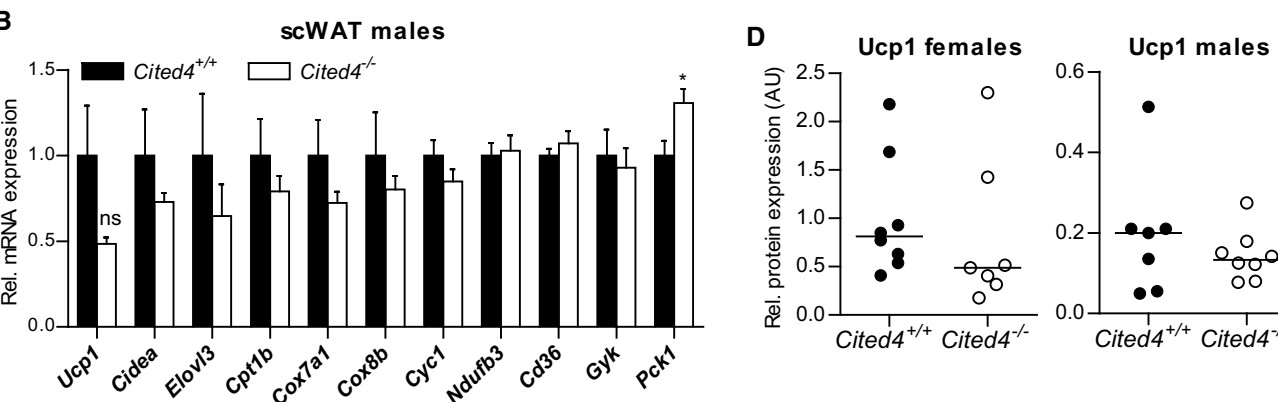

**Figure 6.  Sex-specific involvement of *Cited4* in thermogenic expression upon therapeutic rosiglitazone treatment.**

A, B    mRNA expression in scWAT of mice fed a high-fat diet (HFD) for 11 weeks followed by 5 weeks of HFD with 0.0075% Rosi (qRT–PCR). *t*-test *Cited4*⁻/⁻ vs. *Cited4*⁺/⁺, (A) *n* = 8/7, \*P = 0.029 (*Ucp1*), \*\*P = 0.009 (*Elovl3*), \*\*P = 0.006 (*Cpt1b*), \*\*P = 0.004 (*Cyc1*), \*P = 0.016 (*Ndufb3*), \*\*P = 0.003 (*Cd36*), \*\*P = 0.009 (*Gyk*), (B) *n* = 7/8, \*P = 0.026 (*Pck1*).

C, D    Ucp1 protein expression in scWAT of mice treated as in (A), determined by Western blot with VCP as loading control (*n* = 8/7 for females and *n* = 7/8 for males). Arrow indicates the band specific to Ucp1 (see Appendix Fig S5F ).

Data information: Data presented as mean ± SEM (A and B), individual mice (D, males), or means of replicate blots with independent lysates from the same mice (D, females).
Source data are available online for this figure.

*Cited4* functions outside adipose tissue in the absence of adipocyte progenitor-specific *Cited4* inactivation. Nevertheless, the effects of *Cited4* deficiency on energy expenditure are in support of the importance of adipose tissue *Cited4* function. *Cited4*⁻/⁻ mice had reduced energy expenditure only under Rosi treatment and showed reduced maximal β3-agonist-stimulated oxygen consumption, which is generally attributed to adipose tissue thermogenesis. Furthermore, the effects on energy expenditure were absent in male *Cited4* knockout mice, in which insulin sensitization was intact.

The associations of impaired scWAT function with effects on energy balance and insulin sensitivity in Rosi-treated *Cited4*-knockout mice are consistent with previous reports demonstrating the key role of subcutaneous fat for the therapeutic action of TZDs. This was evident in humans and rodents both at the level of gene expression and browning and at the level of tissue metabolism (Boden *et al*, 2005; Bogacka *et al*, 2005; Festuccia *et al*, 2009; Soccio *et al*, 2017). Obesity is associated with reduced capacity or compromised mitochondrial oxidation in adipose tissue and the ability of TZDs to restore these defects is likely to be important for insulin sensitization, possibly through the improvement of systemic lipid

metabolism (Wilson-Fritch *et al*, 2004; Rong *et al*, 2007; Soccio *et al*, 2017). To which extent the effects of TZDs on Ucp1 expression and uncoupled respiration are contributing to therapeutic action remains to be formally proven. Although TZDs do not induce thermogenesis *per se*, PPARg agonism has been shown to increase the capacity for adipose tissue thermogenesis (Sell *et al*, 2004). In *Cited4*-knockout mice, the reduction in energy expenditure under Rosi treatment was modest, but of similar magnitude to mouse models with specific defects in scWAT browning (Cohen *et al*, 2014).

*Cited4* deficiency did not affect thermogenic expression in scWAT under conditions of prolonged exposure to cold or β3-adrenoreceptor agonist. This implies the existence of different modes of regulation for the induction of thermogenic expression in white fat depending on the stimulus. In line with this notion, the Farmer lab demonstrated the emergence of Ucp1-expressing adipocytes with distinct nature upon β-adrenergic versus Rosi/roscovitine stimulation (Wang *et al*, 2016). The observed differences may be due to different routes of recruitment of the thermogenic adipocytes, i.e., *de novo* differentiation from progenitor cells versus conversion or activation of pre-existing adipocytes. In any case, we propose that

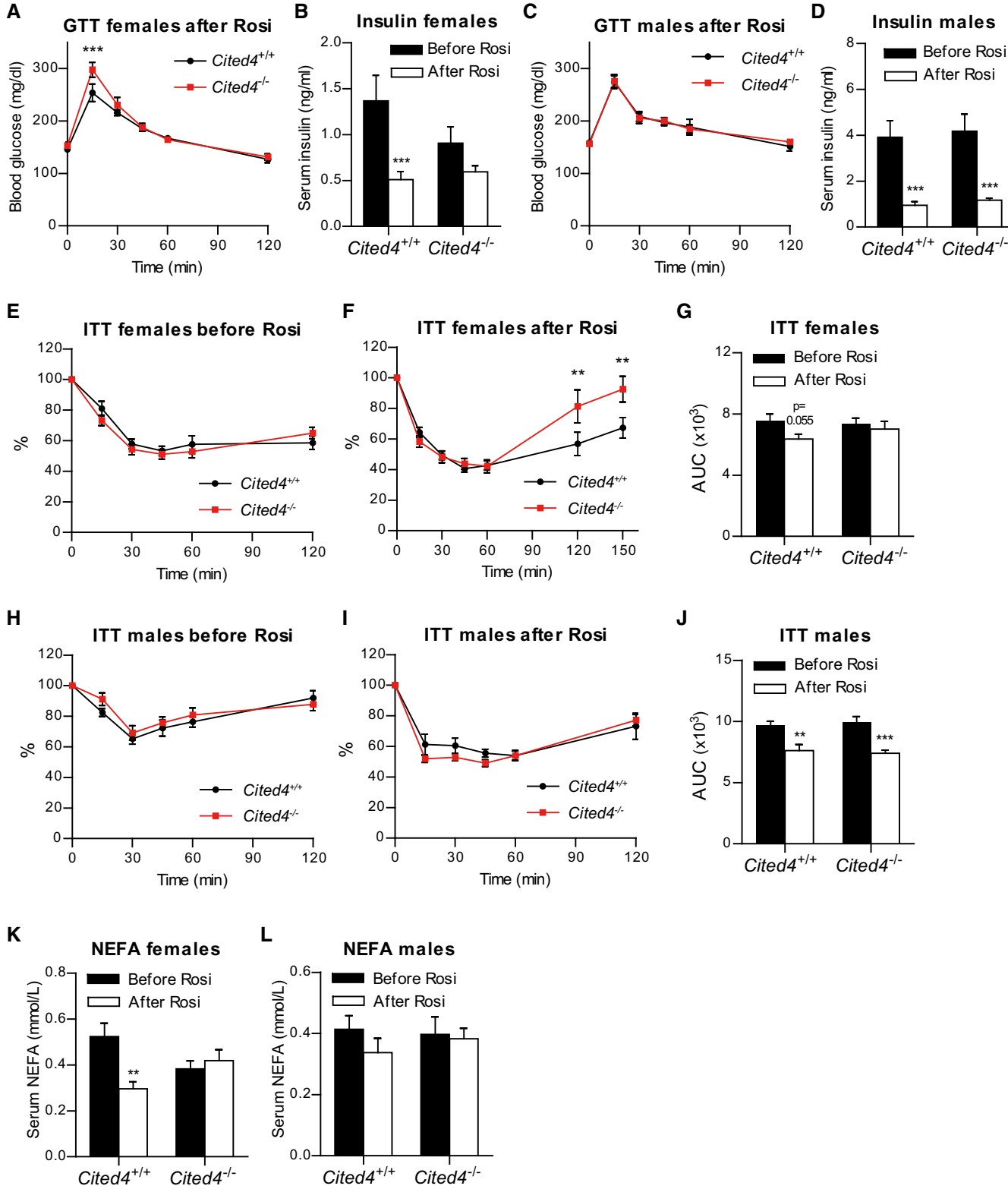

**Figure 7.**

Cited4 is involved in Rosi-mediated beige adipocyte differentiation from progenitors, as concluded from its expression pattern and the differential phenotypes upon inactivation in immature versus committed cells. Is *Cited4* important for white adipocyte formation? Although an effect on the number of lipid-containing cells was observed *ex vivo, Cited4* deficiency did not affect general adipogenic

**Figure 7. Sex-specific involvement of *Cited4* in insulin sensitization upon therapeutic rosiglitazone treatment.**

A    Blood glucose during glucose tolerance test (GTT) on female mice after 11 weeks of HFD and 3 weeks of HFD + Rosi ($n = 8$). ***$P = 0.001$, in repeated-measures $2 \times 2$ ANOVA with Holm–Sidak posttest (*Cited4*$^{-/-}$ vs. *Cited4*$^{+/+}$).

B    Fasting serum insulin in female mice after 10 weeks of HFD ("Before Rosi") or 11 weeks of HFD and 2.5 weeks of HFD+Rosi ("After Rosi") ($n = 8$). ***$P = 0.0009$ in repeated-measures $2 \times 2$ ANOVA with Holm–Sidak posttest (After Rosi vs. Before Rosi); $P = 0.725$ (*Cited4*$^{-/-}$ vs. *Cited4*$^{+/+}$ Before Rosi).

C    Blood glucose during GTT on male mice as in (A) ($n = 5/8$). Repeated-measures $2 \times 2$ ANOVA with Holm–Sidak posttest (*Cited4*$^{-/-}$ vs. *Cited4*$^{+/+}$).

D    Fasting serum insulin in male mice as in (C) ($n = 7/7/8/8$). ***$P = 0.001$ (*Cited4*$^{+/+}$), ***$P = 0.0008$ (*Cited4*$^{-/-}$) in repeated-measures $2 \times 2$ ANOVA with Holm–Sidak posttest (After Rosi vs. Before Rosi).

E, F    Blood glucose during insulin tolerance test (ITT) with 1 U insulin per kg body mass on female mice after 12 weeks of HFD ($n = 8$) (E) or 16 weeks of HFD and 4 weeks of HFD + Rosi ($n = 7$) (F), expressed as % of the 0-time point value. **$P = 0.003$ in repeated-measures $2 \times 2$ ANOVA with Holm–Sidak posttest (*Cited4*$^{-/-}$ vs. *Cited4*$^{+/+}$).

G    Area under the curve (AUC) of blood glucose during the ITT (0–120 min) in (E,F). $2 \times 2$ ANOVA with Holm–Sidak posttest (After Rosi vs. Before Rosi).

H, I    Blood glucose during insulin tolerance test (ITT) on male mice as in (E, F), $n = 8$ (H), $n = 7/8$ (I). Repeated-measures $2 \times 2$ ANOVA with Holm–Sidak posttests (*Cited4*$^{-/-}$ vs. *Cited4*$^{+/+}$).

J    Area under the curve (AUC) of blood glucose during the ITT in (H, J). **$P = 0.001$, ***$P = 0.00006$ in $2 \times 2$ ANOVA with Holm–Sidak posttests (After Rosi vs. Before Rosi).

K, L    Fasting serum non-esterified fatty acids (NEFA) in female (K) or male (L) mice after 10 weeks of HFD ("Before Rosi") or 11 weeks of HFD and 2.5 weeks of HFD + Rosi ("After Rosi"), $n = 7/8/8/8$ (K), $n = 6/6/6/8$ (L). **$P = 0.001$ in $2 \times 2$ ANOVA with Holm–Sidak posttests (After Rosi vs. Before Rosi).

Data information: Data presented as mean ± SEM.

gene expression or the number of adipocytes upon Rosi treatment *in vivo*. Importantly, *Cited4* knockout mice showed indistinguishable fat accumulation upon HFD or Rosi treatment, both potent inducers of adipogenesis. It is noteworthy though that in human SVF cells, mRNA expression of general adipogenic markers was affected to a small degree upon *CITED4* knockdown. A further discordance between the human and mouse cell phenotypes was the sex-independent reduction of *UCP1* expression in human cells with *CITED4* knockdown. How this applies to broader human populations and TZD-treated patients remains to be determined, but it may relate to the strong dependency of human cells on PPARg agonists for adipocyte differentiation in culture.

In the mouse, Cited4 deficiency predominantly or exclusively affected female scWAT. Male *Cited4*$^{-/-}$ mice showed decreases in thermogenic marker gene expression in the steady state and trends of reduction under Rosi treatment. However, the relevance of these effects is questionable since (i) Ucp1 protein could not be detected in scWAT in steady state and (ii) there was no association with reduced energy expenditure or insulin sensitivity under Rosi, in contrast to females. It is noteworthy that the inguinal scWAT depot examined here contains a mammary gland and therefore has high remodeling potential in females. In general, sex differences are characteristic for the subcutaneous fat depot including the differential regulation of progenitor cells (Fried *et al*, 2015; Jeffery *et al*, 2016), but little is known in relation to browning and the TZD response (Benz *et al*, 2012; Fried *et al*, 2015). Estrogen signaling has been implicated in the regulation of thermogenic adipose tissue expression with both central nervous system and local effects in adipocyte progenitor cells (Lapid *et al*, 2014; Martinez de Morentin *et al*, 2014). For instance, it was shown that the induction of Ucp1 by CL-316,243 in mouse gWAT was compromised by experimental ovarian failure (Kim *et al*, 2016). Interestingly, the estrogen receptor is able to interfere with the activity of PPARg and thereby influence PPARg-dependent metabolic processes (Foryst-Ludwig *et al*, 2008; Benz *et al*, 2012). In this light, a functional or physical interaction of Cited4 with estrogen receptors is conceivable. Cited4 and both additional members of the *Cited* family, Cited1 and Cited2, have been shown to act as coactivators of the estrogen receptor (Yahata *et al*, 2001; Lau *et al*, 2013). Notably, *Cited1* has been established as a beige adipocyte marker and is induced by Rosi during progenitor differentiation, which raises the possibility of functional

compensation by *Cited1* in models of *Cited4* gene inactivation (Sharp *et al*, 2012). In any case, the molecular link of Cited4 to the PPARg/TZD transcriptional program could be mediated by direct modulation of PPARg or its cofactors, since this has been observed for Cited2 (Tien *et al*, 2004; Gonzalez *et al*, 2008; Sakai *et al*, 2012).

How Cited4 regulates the TZD response of scWAT sex-specifically at the molecular level and what the implications are for sex-dependent therapeutic efficacy of TZDs in patients remains to be determined. Uncovering how context-specific differential regulation of distinct TZD responses is mediated by Cited4 may open up new possibilities for exploiting and personalizing PPARg modulation in type 2 diabetes or prediabetes.

# Materials and Methods

### Mice

Animal handling and experimentation were performed in accordance with the European Union directives and the German animal welfare act (Tierschutzgesetz). All procedures were approved by local authorities (Regierungspräsidium Karlsruhe). Seven- to twelve-week-old mice were housed in an environmentally controlled room at 22°C on 12-h light/dark cycle and given *ad libitum* access to food and water. Unless otherwise indicated, mice were fed standard chow diet (Art. No. 3437, Provini Kliba AG, Kaiseraugst, Switzerland). Mouse strains used included C57BL/6N, NMRI (Charles River WIGA GmbH, Sulzfeld, Germany), *Cited4* conditional knockout (*Cited4*$^{F/F}$), and *Cited4* constitutive knockout (*Cited4*$^{-/-}$) mice. The *Cited4*$^{F/F}$ line was generated by TaconicArtemis (now Taconic Biosciences, Cologne, Germany) and the GSF (now Helmholtz Zentrum München, Germany). Briefly, the mouse *Cited4* ORF was subcloned using RP23 BAC library and recloned into the basic targeting vector harboring a neomycin selection cassette flanked by flp sites, a thymidine kinase selection cassette as well as two loxP sites enclosing the *Cited4* ORF and the Neo cassette. Targeting was performed in C57BL/6N ES cells and confirmed by Southern blotting. Chimeric mice resulting from blastocyst injection were mated with C57BL/6N Tg(ACTB-Flpe) mice to obtain transgenic founders with excised Neo cassette. *Cited4*$^{+/F}$ mice were crossed to Gt(ROSA)26Sor$^{tm16(cre)Arte}$ mice to obtain the *Cited4*$^-$ null allele. Genotyping was performed by

PCR with the following primers: 5′-AAGATCCAGGCAGCCCTAGC-3′ (Oligo 1; for the detection of the floxed allele) or 5′-TAACCACTGC CAAACGATGG-3′ (Oligo 3; for the detection of knockout allele) (forward) and 5′-CCAACTAGCTGAACCTATTCC-3′ (Oligo 2; reverse). The WT allele generated a band at 359 bp and the floxed allele at 478 bp (Oligos 1/2), whereas the knockout allele was detected at 389 bp (Oligos 3/2; Appendix Fig S1A).

Rosiglitazone treatment was performed by feeding D12450B control diet or D12492 high-fat diet (60% kcal from fat) each supplemented with 0.0075% rosiglitazone (Biomol/Cayman, Hamburg, Germany) resulting in an estimated daily dose of 8–9 mg/kg body weight. The diets incl. the non-supplemented ones were manufactured by Research Diets, New Brunswick, NJ. CL-316,243 (Tocris Bioscience, Bristol, UK) was administered by subcutaneously implanted ALZET® minipumps (DURECT Corporation, Cupertino, CA) at 1 mg/kg per day for 10 days. Body composition was analyzed by NMR using the EchoMRI analyzer (EchoMRI, Echo Medical Systems, Houston, TX). Oxygen consumption (indirect calorimetry), food intake, and locomotor activity measurements were performed with individually housed mice at 22°C in the PhenoMaster Home Cage System (TSE system, Bad Homburg, Germany). Intraperitoneal glucose tolerance test (GTT) was conducted by injecting the mice with 2 g glucose (Sigma-Aldrich, Munich, Germany) per kg body weight after a 5-h fast. Insulin tolerance test (ITT) was performed by injecting intraperitoneally 1 U insulin (HUMINSULIN®; Lilly, Bad Homburg, Germany) per kg body mass or lean mass as indicated after a 5-h fast. Blood glucose was measured with an Accu-Check glucometer (Roche Diagnostics, Mannheim, Germany). Non-esterified fatty acids (NEFA) and insulin were determined in serum samples using the NEFA-HR (2) (Wako Diagnostics, Neuss, Germany) and the ALPCO Insulin ELISA (BioCat, Heidelberg, Germany) kits, respectively.

**Assessment of mouse cardiovascular function**

Mice were acclimatized to the housing conditions for 4 weeks prior to analysis. The basal cardiovascular functions were assessed by measuring tail-cuff blood pressure in conscious mice (at 9 weeks of age) using the MC4000 Blood Pressure Analysis Systems (Hatteras Instruments Inc., Cary, North Carolina, USA). Prewarmed metal platforms in metal boxes were used to restrain the animals. The tails were fixed through a tail-cuff in a notch with an optical path containing an LED light and a photosensor. The blood pulse wave in the tail artery was determined as an optical pulse signal. The system software allowed automated detection of pulse, cuff inflation, and pressure evaluation. Following five initial inflation runs for acclimatization, 12 measurement runs were conducted per animal in one session. Runs with movement artifacts were not included in the analysis. The measurements were taken over four consecutive days (between 8:30 and 11:30 AM) after a day of training for the adaptation of animals to the protocol. Additionally, the heart weight was determined at 15 weeks of age along with body weight and tibia length.

**Histology**

Dissected tissue was rinsed in PBS, fixed in 4% paraformaldehyde at room temperature for 24 h, embedded in paraffin, cut into series of 5-μm-thick sections, and mounted onto glass slides. The section was deparaffinized (xylene), rehydrated (ethanol dilutions), and subjected to antigen retrieval in citrate buffer (pH 6.0) through boiling at 95°C for 20 min. The slides were blocked with 2% BSA in PBS, for 1 h at room temperature followed by overnight incubation at 4°C with rabbit α-caveolin-1 (Cav-1) polyclonal antibody 1:400 in 2% BSA in PBS (Cell Signaling, Danvers, MA). Following washes with PBS, goat α-rabbit IgG-Alexa Fluor® 488 secondary antibody at 1:400 dilution in 2% BSA/PBS (Thermo Fisher Scientific, Rockford, IL) was applied for 60 min at room temperature. The sections were washed and mounted with ProLong® Gold Antifade Reagent (Cell Signaling, Danvers, MA) and coverslips. Three to five images per mouse were acquired at 100× magnification using the Zeiss Cell Observer with identical settings and acquisition times. For the calculation of cell surface, images were analyzed with a Fiji software algorithm (ImageJ). Cell mass was calculated from the median cell area per mouse assuming spherical cell shape and adipocyte density of 0.96 g/ml. The number of adipocytes per mg tissue was estimated as 1/(median cell mass) and multiplied by tissue mass to obtain the total number of adipocytes per depot.

**SVF preparation from human and mouse adipose tissue**

Biopsies from human abdominal subcutaneous fat were collected during bariatric surgery at the Department of Surgery, University Hospital Heidelberg, with approval by the Institutional Review Board of the Medical Faculty of the University of Heidelberg in accordance with the Declaration of Helsinki and its later amendments. Preoperative informed consent was obtained from all patients for the use of samples. Inguinal/gluteal subcutaneous adipose tissue was excised from 7- to 8-week-old mice. Murine or human fat biopsies were minced with scissors and digested in Hank's balanced salt solution (HBSS; Sigma-Aldrich, Munich, Germany) containing 0.1 w.u./ml purified collagenase (LS005273; Worthington Biochemical, Lakewood, NJ), 2.4 U/ml purified neutral protease (LS02104; Worthington Biochemical), 4 mM $CaCl_2$, and 0.05 mg/ml DNase I (1284932001; Roche Diagnostics, Grenzach-Wyhlen, Germany) for 50–60 min at 37°C in a shaker at 70 rpm. The suspensions were strained through a 300-μm mesh (4-1411; Neolab, Heidelberg, Germany) and centrifuged at 145 g for 10 min at 20°C to separate the SVF and mature adipocytes. The mature adipocyte fraction was discarded, unless otherwise indicated, and SVF cells were washed in BSA buffer (0.5% BSA, 1 mM EDTA in D-PBS) and collected by centrifuging at 300 g for 5 min at 20°C.

**Isolation of primary adipose progenitors by MACS®**

After obtaining the SVF, adipose progenitors were isolated as described previously (Bayindir et al, 2015; Babaei et al, 2017). In brief, the SVF cell pellet was resuspended in the appropriate volume of BSA buffer and preincubated with FcBlock (anti-CD16/32; eBioscience, Frankfurt, Germany) for 10 minutes on ice. Cells were then stained with biotin-conjugated lineage antibodies, namely Ter119 (TER-119), CD31 (390), and CD45 (30-F11) antibodies (eBioscience), for 30 min on ice to label erythrocytes, endothelial cells, and hematopoietic cells, respectively. After staining, cells were washed and incubated with Streptavidin MicroBeads (130-048-102,

Miltenyi Biotec, Bergisch Gladbach, Germany) for magnetic separation with an OctoMACS Separator according to the manufacturer's instructions. Ter119⁻CD31⁻CD45⁻ (Lin⁻) cells in the flow-through were incubated with Anti-Sca-1 MicroBeads (130-106-641, Miltenyi Biotec) for magnetic separation following the manufacturer's protocol. The Lin⁻Sca1⁺ cells retained in the column were collected by removing the column from the magnet and plunging through with 1 ml BSA buffer. The cells were then centrifuged at 300 $g$ for 5 min at 4°C for further processing.

## Isolation of primary cell populations from mouse adipose tissue by FACS

After obtaining SVF single cell suspensions, different cell populations were sorted as described previously (Bayindir *et al*, 2015). Cells were preincubated with FcBlock (anti-CD16/32; eBioscience) for 10 min on ice. Erythrocytes were depleted via magnetic separation with an OctoMACS Separator according to the manufacturer's instructions, following incubation with Anti-Ter119 MicroBeads (130-049-901, Miltenyi Biotec) for 15 min on ice. The flow-through was stained with CD45-FITC (30-F11, eBioscience), CD31-eFluor450® (390, eBioscience), CD29-PerCP-eFluor® 710 (HMb1-1, eBioscience), CD34-Alexa Fluor® 647 (RAM34, BD Biosciences, Heidelberg, Germany), and Sca-1-Alexa Fluor® 700 (D7, eBioscience) for 30 min on ice. Cells were washed, sorted as Lin(CD31/CD45)⁻CD29⁺CD34⁺Sca1⁺ with a BD FACS Aria (BD Biosciences), and centrifuged at 300 $g$ for 5 min at 4°C and resuspended either in culture medium with bFGF for culturing or in QIAzol® Lysis Reagent (Qiagen, Hilden, Germany) for RNA isolation. FCS files exported via BD FACSDiva™ software were analyzed with FlowJo Software (FlowJo, Ashland, OR).

## Primary cell culture and adipogenic differentiation

Stromal vascular fraction cells derived from human fat biopsies were plated in culture medium (DMEM, 10% FCS, 1% penicillin/streptomycin [Life Technologies™, Darmstadt, Germany]) supplemented with 10 ng/ml recombinant human bFGF (R&D Systems) for up to three passages. Primary adipose progenitors isolated from murine inguinal subcutaneous depot by MACS® or FACS were seeded out on BIOCOAT laminin-coated plates in culture medium supplemented with 10 ng/ml recombinant murine bFGF (R&D Systems) at a density of 1–3 × 10⁴ cells/cm². At confluency, adipogenic differentiation was induced with DMEM, 10% FCS (or 10% NCS for human SVF), 1% penicillin/streptomycin, 1 μg/ml insulin, 500 nM dexamethasone, 3 nM 3,3,5-triiodo-L-thyronine (T3), 0.5 mM 3-isobutyl-1-methylxanthine (IBMX; human cells only) [Sigma-Aldrich]) for 2 days (or 3 days for human SVF). Induction was followed by DMEM supplemented with 5% FCS (or 5% NCS for human SVF), 1% penicillin/streptomycin, 1 μg/ml insulin and 3 nM T3) for up to 6 days (or 11 days for human SVF). Wherever indicated, cPGI₂ (Biomol, Hamburg, Germany) was added to the medium at 1 μM for up to 8 days, rosiglitazone (Rosi; Biomol) was added to the medium at 100 nM for up to 5 days (mouse) or up to 14 days (human) and pioglitazone (Pio; Biomol) was added at the indicated concentration for up to 5 days. Corresponding concentrations of ethanol and DMSO served as control for cPGI₂ and Rosi or Pio, respectively.

Human SVF cells were transfected at 60–70% confluency with Silencer® Select small interfering RNAs (siRNA) targeting *CITED4* (s225785 and s46475; Life Technologies) or an equimolar amount of negative control (No. 1; 4390843; Life Technologies) using Lipofectamine® RNAiMAX transfection reagent (Life Technologies). Lipofectamine® RNAiMAX in serum-free Opti-MEM (Life Technologies) medium (20 μl/ml) was mixed with Opti-MEM containing siRNA and incubated at room temperature (RT) for 5 min before transfer to the cultures with medium at 5 nM final concentration.

For the transfection of primary Lin⁻Sca1⁺ cells from *Cited4*^F/F mice with StemMACS™ Cre recombinase (130-101-113; Miltenyi Biotec) or StemMACS™ Nuclear eGFP messenger RNA (mRNA) (130-101-119; Miltenyi Biotec), a mixture of Lipofectamine® RNAiMAX transfection reagent (2 μl/ml) and Opti-MEM medium was combined with Opti-MEM containing the respective mRNA (500 ng/ml final concentration). After incubation at RT for 20 min, the transfection complexes were added dropwise to the cells for overnight incubation either before or 3 days after differentiation induction. To determine transfection efficiency, the cells were collected by trypsinization and analyzed for GFP expression by flow cytometry using a BD FACS Calibur (BD Biosciences).

## C3H10T1/2 and 3T3-L1 culture and adipogenic differentiation

C3H10T1/2 cells (Clone 8, ATCC® CCL-226™) were propagated in DMEM containing 10% FCS and 1% penicillin/streptomycin until confluency. Adipogenic differentiation was induced by medium supplemented with 0.25 μM dexamethasone, 0.5 mM IBMX, 1 μg/ml insulin, and 3 nM T3 for 4 days, with a medium change every 2 days. Induction was followed by differentiation maturation medium (culture medium with 1 μg/ml insulin and 3 nM T3) for 6 days. 1 μM rosiglitazone or the corresponding concentration of DMSO was added to the medium throughout the differentiation. 3T3-L1 preadipocytes (ATCC® CL-173™) were cultured in low (1 g/l)-glucose DMEM containing 10% FCS and 1% penicillin/streptomycin until confluency. Adipogenic differentiation was induced with high (4.5 g/l)-glucose DMEM containing 10% FCS and 1% penicillin/streptomycin supplemented with 0.5 μM dexamethasone, 0.5 mM IBMX, 1 μg/ml insulin for 4 days. Following induction, cells were cultured in high (4.5 g/l)-glucose DMEM with 10% FCS, 1% penicillin/streptomycin, 1 μg/ml insulin for an additional 6 days and further 2 days without insulin. 1 μM rosiglitazone or the corresponding concentration of DMSO was added to the medium up to day 6.

## Cellular respiration assay (Seahorse)

Lin⁻Sca1⁺ cells isolated from the scWAT of female mice were seeded into XF96 V3-PS cell culture microplate (Seahorse Bioscience, Copenhagen, Denmark), cultured, and differentiated as above. Insulin and T3 were omitted from the media at day 7 of differentiation. At day 8 of differentiation, Mito Stress Test was performed following the manufacturer's instructions. Briefly, the medium was replaced with prewarmed, unbuffered Seahorse Assay Medium (DMEM basal medium [Sigma Cat. No. D5030] supplemented with 25 mM glucose, 1 mM pyruvate, 1 mM glutamine, 15 mg/l phenol red, pH 7.4), and cells were incubated at 37°C in a

$CO_2$-free incubator for at least 1 h. After calibration, respiration was measured in an XF96 Extracellular Flux Analyzer (Seahorse Bioscience). Cells were sequentially treated at the indicated time points with oligomycin (4 μM), CL-316,243 (1 μM), trifluorocarbonylcyanide phenylhydrazone (FCCP, 2 μM), and rotenone-antimycin A (rotenone-AA, 1 μM each). Oxygen consumption rates were determined at the indicated time points and extracted by the Seahorse XF-96 software. After completion of the assay, DNA content was determined using the CyQUANT Cell Proliferation Assay Kit (Thermo Fisher Scientific) according to the manufacturer's instructions. In brief, 200 μl of CyQUANT 1× lysis buffer with 1× CyQUANT GR dye was added per well while maintaining the plate on ice. The lysates were transferred to a flat bottom 96-well plate along with a standard dilution series. Fluorescence was measured on a Mithras LB 940 plate reader (Berthold, Bad Wildbad, Germany) at 485/535 nm. DNA content was calculated using http://www.elisaanalysis.com/app and 5-Parameter Logistic Regression. Oxygen consumption rates were normalized to total DNA content per well.

### Lipid staining by LipidTOX™

*Ex vivo* differentiated adipocytes were washed and fixed with 4% paraformaldehyde solution (Thermo Fisher Scientific, Schwerte, Germany) for 15 min at RT. After washing with 1× PBS (Life Technologies), blocking was performed by incubating the cells in 5% BSA (Sigma-Aldrich) in 1× PBS for 1 hour at RT. For LipidTOX™ (Life Technologies) and 4,6-diamidino-2-phenylindole (DAPI; Sigma-Aldrich) co-staining, the staining solution was prepared with 1:200 LipidTOX™ Red Neutral Lipid stain and 0.5 μg/ml DAPI in 1× PBS. Cells were stained for minimum 30 min at RT in the dark prior to imaging.

A Cell Observer Z1 microscope and ZEN software (Carl Zeiss, Oberkochen, Germany) were used for fluorescence imaging. Forty images/group captured randomly across the wells and analyzed with an ImageJ (http://rsbweb.nih.gov/ij) algorithm. In brief, nuclei were segmented and counted and the LipidTOX™ signal intensity was quantified. The signal intensity was then used to count the LipidTOX[+] cells by setting a threshold for the mean signal intensity for all samples. The number of LipidTOX[+] cells was depicted as the percentage of DAPI[+] nuclei per field.

### RNA isolation, cDNA synthesis, and quantitative real-time polymerase chain reaction

RNA was isolated from cells or snap-frozen pulverized tissue with QIAzol® reagent and the RNeasy® Micro/Mini Kit (Qiagen) following the manufacturer's protocol. Complementary DNA (cDNA) synthesis was performed with 200–1,000 ng total RNA using QuantiTect® Reverse Transcription Kit (Qiagen). Quantitative real-time polymerase chain reaction (qRT–PCR) was performed using TaqMan® Gene Expression Assays (Life Technologies) and TaqMan® Gene Expression Master Mix on a StepOnePlus™ Real-Time PCR System (Life Technologies). Normalization to *Tbp/TBP* (TATA box binding protein) and calculation of relative expression values were performed with the ($\Delta\Delta C_T$) method. 18S ribosomal RNA was used instead of *Tbp* for normalization of *Cited4* values in gene expression analysis comparing different tissues.

### Microarray expression profiling

Expression profiling was performed with total RNA and the GeneChip® Mouse Gene 2.0 ST array platform (Affymetrix, High Wycombe, UK) by the DKFZ Genomics and Proteomics Core Facility according to the manufacturer's instructions. Gene-level intensity values were calculated from the CEL files using robust multiarray average (RMA) normalization with the Affymetrix Expression Console and MoGene-2_0-st-v1 library files and annotation files. Ranking of probe sets (gene level) for differential expression was based on *P*-values obtained from *t*-test with Welch's correction performed with TM4: MultiExperiment Viewer (MeV) software (Saeed *et al*, 2003). Gene set enrichment analysis (GSEA; Subramanian *et al*, 2005) was performed on the complete probe dataset using the c2.cp.kegg.v4.0.symbols.gmt and the c3.tft.v5.0.symbols.gmt gene set collections (Molecular Signatures Database, MSigDB; http://www.broadinstitute.org/gsea/msigdb/index.jsp). The parameters of the analysis were adjusted as follows: permutation type = phenotype (1,000 × ); enrichment statistic = weighted; metric for ranking genes = Signal2Noise; normalization mode = meandiv. Ranking of gene sets was by the false discovery rate (FDR).

### Immunoblotting

For protein extraction, primary adipocytes were lysed in RIPA buffer (150 mM NaCl, 1% Triton™ X-100, 50 mM Tris pH 8.0, 0.5% sodium deoxycholate, 0.1% SDS, 1× cOmplete™ protease inhibitor cocktail tablet [Roche]). After incubation on ice for 30 min, the lysates were centrifuged at 13,000 rpm, 4°C for 15 min. Snap-frozen pulverized adipose tissue was lysed in 50 mM Tris pH 7.4, 1 mM EDTA pH 8, 1.5 mM $MgCl_2$, 10 mM NaF, 2 mM $Na_3VO_4$, 1 mM DTT, 1× cOmplete™ protease inhibitor cocktail tablet, using a TissueLyzer. After 1 h on ice, the lysates were centrifuged shortly and the aqueous phase was supplemented with 150 mM NaCl, 0.5% deoxycholic acid, 1% NP-40, 0.1% SDS, 1% glycerol and shaken at maximum speed at 4°C for 1 h. The lysates were cleared by centrifugation at 13,000 rpm, 4°C for 30 min. Protein concentration was determined using the Pierce® BCA Protein Assay Kit (Thermo Fisher Scientific). After SDS–PAGE and wet blotting, the nitrocellulose membranes were blocked either in 5% BSA or in 5% skim milk in TBS-0.1% Tween-20 for 1 h at RT. The membranes were incubated with antibodies detecting Ucp1 (1:1,000; PA1-24894; Thermo Fisher Scientific), Cited4 (1:500; ab105797; Abcam), and Vcp (1:2,000; ab11433; Abcam) overnight at 4°C. Subsequently, they were incubated with the secondary antibodies goat anti-rabbit IgG (H + L)-HRP (1:4,000; 172-1019; Bio-Rad, Munich, Germany) or anti-mouse IgG (H + L)-HRP (1:5,000; 170-6516; Bio-Rad) for 1 h at RT. Following the enhanced chemiluminescence system reaction (ECL™ Western Blotting Detection Reagents; GE Healthcare, Solingen, Germany) protein bands were detected and quantified using the ChemiDoc™ XRS[+] Molecular Imager® with ImageLab™ Software (Bio-Rad).

### Statistical analysis

For mouse experiments, a sample size calculation was performed by a statistician. For cell culture experiments in general, sample

### The paper explained

#### Problem

Glitazones represent an exception among antidiabetic drugs in that they target excess substrate metabolism and insulin sensitivity as pathogenic causes in the metabolic syndrome and type 2 diabetes as opposed to treating disease symptoms. However, they are rarely prescribed due to their side effects. Understanding the mechanism of action of glitazones can help the exploitation of the nuclear receptor PPARg pathway as the main glitazone target to develop safer drugs.

#### Results

We found that the expression of the transcription cofactor gene *Cited4* was upregulated by glitazone treatment in immature progenitor cells derived from adipose tissue, a major target of glitazones. Genetic inactivation of *Cited4* resulted in defective differentiation of progenitors to beige adipocytes as reflected by the reduced induction of uncoupling protein 1 (Ucp1) and other thermogenic adipocyte markers by rosiglitazone in primary cultured human and murine cells as well as in adipose tissue in mice. This phenotype was more penetrant in females and was associated with reduced whole-body energy expenditure, possibly due to lower capacity for adipose tissue thermogenesis in *Cited4*-deficient mice. Furthermore, *Cited4* inactivation resulted in compromised therapeutic insulin sensitization in female but not male mice when rosiglitazone was administered in the context of diet-induced obesity.

#### Impact

Our study revealed unexpected sex-, tissue-, and context-specific aspects of glitazone action which are relevant for the development and personalization of new therapeutic approaches targeting PPARg and adipose tissue metabolism in type 2 diabetes and prediabetes.

## Data availability

Microarray data have been deposited in the ArrayExpress database under accession number E-MTAB-6796.

**Expanded View** for this article is available online.

## Acknowledgements

We thank Dr. Adam J Rose and Dr. Claus Kremoser for valuable discussions and know-how, Dagmar Kindler, Patrick Matei, Jakob El Kholtei, and Anna Taranko for technical support, Dagmar Walter for technical know-how, the DKFZ Genomics and Proteomics Core Facility, Steffen Schmitt and the DKFZ Imaging and Cytometry Core Facility, the DKFZ Center for Preclinical Research. This work was supported by the Human Frontier Science Program (RGY0082/2014), a stipend from the Helmholtz International Graduate School for Cancer Research to I.B.B, a Novo Nordisk postdoctoral fellowship in partnership with Karolinska Institutet to L.K., the German Federal Ministry of Education and Research (Infrafrontier grant 01KX1012), the Deutsche Forschungsgemeinschaft (HE 3260/8-1), and the Brain Tumor Network (NGFNplus #01GS0883).

## Author contributions

Conceptualization, IB-B, AV; Investigations, IB-B, GW, SL, TS, MS, RB, LK, NS; Resources, JG, ATB, BPM-S, PL; Software, DK; Funding Acquisition, IB-B, LK, JLR, MHA, PL, SH, AV; Supervision, MHA, JLR, BPM-S, MH, PL, SH, AV; Writing, IB-B, AV.

## Conflict of interest

The authors declare that they have no conflict of interest.

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
