## [Review Process File · EMBO Molecular Medicine]

Cited4 is a sex-biased mediator of the antidiabetic glitazone response in adipocyte progenitors

Irem Bayindir-Buchhalter, Gretchen Wolff, Sarah Lerch, Tjeerd Sijmonsma, Maximilian Schuster, Jan Gronych, Adrian T Billeter, Rohollah Babaei, Damir Kronic, Lars Ketscher, Nadine Spielmann, Martin Hrabe de Angelis, Jorge L Ruas, Beat P Müller-Stich, Mathias Heikenwalder, Peter Lichter, Stephan Herzig, Alexandros Vegiopoulos

Review timeline:

Submission date:	23 October 2017
Editorial Decision:	27 November 2017
Revision received:	27 April 2018
Editorial Decision:	24 May 2018
Revision received:	07 June 2018
Accepted:	12 June 2018

Editor: Céline Carret

Transaction Report:

1st Editorial Decision

27 November 2017

Thank you for the submission of your manuscript to EMBO Molecular Medicine and for your patience during the review process. We have now heard back from the three referees whom we asked to evaluate your manuscript.

You will see from the comments pasted below that they all found the study of interest, but at this stage not sufficiently developed to reach publication level. While ref. 1 feels that the results are over-stated and more mechanism must be provided, ref. 2 & 3 are more supportive, even though overall the same criticisms are made. Overall, We would like to encourage you to provide more experiments to corroborate your findings, but also better controls, and better data presentation in order to develop the gender difference insights, as this is a very interesting point of the study. Very detailed suggestions are provided that really would improve the conclusiveness and clinical relevance if followed. I'd like to note that while ref. 2 agrees with ref. 1 that a conditional mouse KO for *cited4* would be ideal, still this referee considers this beyond the cope of this paper and ref. 3, during our cross-commenting exercise, mentioned that it would not be realistic in this context.

We would welcome the submission of a revised version within three months for further consideration and would like to encourage you to address all the criticisms raised as suggested to improve conclusiveness and clarity. Please note that EMBO Molecular Medicine strongly supports a single round of revision and that, as acceptance or rejection of the manuscript will depend on another round of review, your responses should be as complete as possible.

Please also contact us as soon as possible if similar work is published elsewhere. If other work is published we may not be able to extend the revision period beyond three months, should you need it.

I look forward to receiving your revised manuscript.

***** Reviewer's comments *****

Referee #1 (Comments on Novelty/Model System for Author):

This concern is described in my comments to the authors

Referee #1 (Remarks for Author):

In the present work, the authors describe the participation of the cofactor Cited4 in the modulation of adipocyte differentiation from adipocyte progenitors upon rosiglitazone stimulation. The authors claim that this cofactor is involved in the rosiglitazone-mediated browning of scWAT. Surprisingly, the observed effects are not only tissue-specific (confined to scWAT), but also gender-restricted (occurring in females only). Unfortunately, the authors fall short in identifying the mechanisms behind the observed phenotype, making the work largely descriptive.

Moreover, many of the claims are not supported by the current data. The beige phenotypes observed *in vitro* from adipocyte progenitors and *in vivo* from subcutaneous fat pads after rosiglitazone treatment are only based on gene expression analysis. Functional analyses should be performed to evaluate if changes found in gene expression translate into enhanced mitochondrial biogenesis, uncoupling, or fatty acid oxidation. Effects on energy expenditure are only investigated in living animals. This is a major concern given the ubiquitous expression of Cited4 and the type of mouse model used (whole body KO). A similar flaw can be formulated for the insulin sensitization experiments presented in the manuscript. It cannot be concluded that the impaired insulin sensitization upon therapeutic rosiglitazone treatment in Cited4^{-/-} mice are mediated by white adipose tissue since Cited4 is expressed in all major metabolic or endocrine organs (brown adipose tissue, skeletal muscle, liver etc...). Animals with white adipose tissue specific deletion must be used to reinforce the authors' conclusions.

Finally, the observation that gender-specific effects of Cited4 contribute to the phenotype is interesting, yet the authors provide little discussion about the possible mechanisms underlying this effect. Altogether, this manuscript leaves many questions unexplained and unexplored.

Referee #2 (Comments on Novelty/Model System for Author):

Overall the work appears well done technically. The novelty of another transcriptional co-regulator in adipocytes is not high, especially given the lack of molecular mechanism. Medical impact of these finding is remote, as much more would need to be elucidated. The most interesting observations (i.e. effects of Cited4 in females not males, subcutaneous but not other fat depots, in response to TZD but not adrenergic stimulation) are purely descriptive.

Referee #2 (Remarks for Author):

Bayindir-Buchhalter et al submit a manuscript "Cited4 is a sex-based mediator of the antidiabetic glitazone response in adipocyte progenitors". They identified Cited4 by global profiling of mouse adipocyte precursors as a transcript induced by the PPAR γ agonist cPGI₂, and showed similarly transient induction during adipogenesis stimulated by rosiglitazone. A Cited4^{-/-} mouse model was developed, and female adipocyte precursors from these mice show a selective defect in expression of beige adipocyte genes (though not general adipocyte markers). This was confirmed in human SVF-derived adipocytes with siRNA knockdown of CITED4, and in inducible Cre-mediated knockout of Cited4 *f/f* mouse cells in culture. Expression profiling of wild type vs knockout adipose progenitors at 2 days of differentiation was consistent with their model that Cited4 is required for maximal expression of genes in beige adipocyte pathways. Whole body knockout mice were also

studied, and while there was no gross difference in adipogenesis, the beiging response of scWAT (particularly Ucp1 mRNA and protein expression) was blunted in female knockout mice. Interestingly, this phenotype was not observed in male mice, nor in gonadal WAT or interscapular BAT, nor when beiging was induced by cold or beta agonists. Rosi-treated female Cited4 knockout mice had a mild defect in energy expenditure despite similar physical activity and food intake, consistent with less uncoupled respiration. Finally, rosi-treated female knockout mice on high fat diet had impaired glucose and insulin tolerance, consistent with insulin resistance despite similar body weights. This was not observed in male knockouts, again indicating a sex-selective requirement for Cited4 in rosi-mediated scWAT beiging.

Overall these are interesting observations and it is a well-written manuscript, but there are a number of issues to address:

1) Figure 1A-B nicely shows transient induction of Cited4 during adipogenesis from precursor cells isolated from mouse adipose tissue. However, given the sex differences described later in the paper, the sex is not clearly described in these figures. Was the Cited4 induction sex-dependent? Also, the mouse 3T3-L1 cell line is very commonly used to study adipocyte differentiation, and it would be valuable to know whether Cited4 is also transiently induced in a rosi-dependent manner in this model as well.

2) Cited4 f/f and -/- mice are described for the first time. The targeting strategy is described somewhat in the methods, but this manuscript would benefit from a supplemental figure detailing the constructs, genotyping, and validation. For instance, I was confused how the entire ORF could be used in the targeting construct until I looked on a genome browser and saw Cited4 is a small single exon gene.

3) Figure S1A and S4A make it clear that Cited4 mRNA is undetectable in SQ fat from -/- mice. For this initial report of the knockout, ideally this should also be demonstrated at the protein level. A brief internet search revealed commercial antibodies (i.e. Anti-CITED4 antibody, AbCam ab105797) that reportedly detect the protein in mouse tissues like heart. Notably, mouse GeneAtlas shows similar Cited transcript abundance in muscle as in white fat, so it would also be valuable to demonstrate loss of transcript and protein in this tissue. (Also, is there a cardiac phenotype to these whole body knockout mice?) It should also be noted in the discussion that the whole body knockout would also result in loss of expression in non-adipose tissues like skeletal muscle, which could be relevant to insulin sensitivity. Tissue-specific and inducible knockouts are beyond the scope of this paper but could be mentioned as future directions. Furthermore, Cited4 is a member of gene family with Cited1-3, and no mention is made of these other members: their expression, regulation, potential functional compensation, etc.

4) Figure 1C looks at a panel of brown/beige genes in the cultured cells, yet omits Elov13 which in later figures is the gene most affected by Cited4 knockout in adipose tissue.

5) Figure 1D shows a marked induction of Cited4 in rosi-treated human SVF-derived adipocytes. However, it is unclear whether this is an effect of rosi or simply adipocyte differentiation. Figure S1C shows that expression of the adipocyte marker adiponectin is ~100X less in the absence of rosi (note log scale), whereas in mouse precursors the difference was only ~5X (Figure 1C). This could reflect the fact that human adipocyte models generally require a TZD for full differentiation, making it challenging to study human cultured adipocytes in the absence of drug. Furthermore, in mouse adipocytes Cited4 expression was transient during adipogenesis and strikingly returned to basal levels in mature adipocytes (Figure 1A-B). Rather than a single day 9 time point, a time course of human adipogenesis would be necessary to show this pattern in human cells.

6) Figure 1E uses two different siRNAs to knock down expression of Cited4, yet the two have different effects on some genes (Adipoq and Slc2a4). The suppression of Adipoq by siCited4.1 is inconsistent with the author's model that Cited4 is specific for "beige" but not "white" adipocyte genes, so this deserves comment.

7) Figure 1I should be on two separate graphs rather than left and right axes, which are confusing.

8) The expression profiling in figure 2 would benefit from more gene level data in addition to the pathway analyses presented. For instance, heat maps, Venn diagrams, and/or volcano plots of regulated genes would add to the analysis, particularly with identification of key genes like *Ucp1*, *Cidea*, *Elovl3*, etc. Also, the focus is entirely on rosi-induced genes, and figures like these would show whether rosi-repressed genes are similarly affected by *Cited4* knockout. Finally, the profiling only at day 2 progenitors may be misleading, as it remains possible that the differences in gene expression do not persist in mature adipocytes.

9) In Figure 3B, in male scWAT it appears that, in the absence of rosi, loss of *Cited4* clearly decreases expression of all 5 "beige" genes (comparing the first two bars). Was this effect significant? If so, it is also inconsistent with the model that *Cited4* specifically affects only rosi-mediated gene expression.

10) In Figure 3A-B, the effects of *Cited4* knockout on *Ucp1* mRNA expression in scWAT are a similar 2-2.5x in both females and males, yet the in 3C the protein effects are markedly different. The reasons for this disconnect between RNA and protein are not discussed. Also, the Western Blots in figure 1C differ between the sexes, with a much fainter band in females along with the appearance of an apparent nonselective band below *Ucp1*. Given this result, it would be valuable to directly compare male and female scWAT on the same blot to investigate the sex difference.

11) Figure S4 in the appendix contains key data that belongs in the main manuscript. Figure S4A shows that *Cited4* mRNA is not rosi-induced in scWAT, which deserves mention given the effect of rosi on cultured cells in Figure 1. The expression data in S4B-C is also very relevant but needs some additional pieces: all three fat depots need to be compared in both sexes, as well as cardiac/skeletal muscle where the RNA is also abundant. Figures S4D and S4E show the key and very interesting negative results (even mentioned in the abstract) that *Cited4* does not affect scWAT beiging by beta3-agonist or cold exposure. Other negative data in main Figure 3 (i.e. 3E and 3F showing no effect of knockout in other depots) could be moved to supplemental to make room for this more relevant data in the main text. Also, figures S4D and S4F would benefit from included the unexposed scWAT as a control, to confirm the extent of beiging by drug and cold (similar to the absence of rosi in the rest of figure 3).

12) Figure 4 shows a significant difference in oxygen consumption in female mice in the presence of rosi, as this is the sex and treatment proposed to be relevant by the authors' model. However, it would be valuable to know whether this difference exists the absence of rosi, or in male mice.

13) Similarly, Figure 5 shows HFD-fed mice only after treatment with rosi, but does not show the effect of rosi treatment. This is particularly relevant for the ITT in Fig 5D, in which there is minimal response to insulin in the control female mice and virtually none in the *Cited4* knockouts. This indicates the mice were extremely insulin resistant yet rosi should have been insulin sensitizing. Including an untreated but HFD-exposed group in this experiment would be necessary to show the insulin sensitizing effect of rosi, and to make a more convincing case that the effect is lost in the *Cited4* knockout mice. Finally, given how important the insulin resistance of the knockout mice is to the overall conclusion of the manuscript, the gold standard assay of insulin sensitivity (hyperinsulinemic-euglycemic clamp) would be a much better test than an ITT.

14) Finally, no mechanism is explored or even proposed for *Cited4*. The gene name "Cbp/p300 interacting transactivator with Glu/Asp rich carboxy-terminal domain 4" indicates that it is itself a transcriptional co-regulator, so could its effects on gene regulation be direct as part of transcriptional regulatory complex with PPARgamma? There is also literature (briefly mentioned in the introduction) about *Cited4* and C/EBP transcription factors in cardiac hypertrophy, yet C/EBPs are also highly relevant in adipogenic gene regulation. Even if exploration of the molecular mechanism of *Cited4* is beyond the scope of this paper, and brief review of *Cited4* known biology and its potential role in adipocytes should be discussed.

Referee #3 (Remarks for Author):

Bayindir-Buchhalter et al. investigated the role of the PPAR γ transcriptional cofactor *Cited4* as a target and mediator of rosiglitazone in adipocyte progenitor cells. They report that *Cited4* is required

for rosiglitazone-mediated induction of thermogenic expression in subcutaneous fat mainly in female mice. Importantly, *Cited4* appears not to be involved in beta-adrenergically or cold stimulated browning.

Overall, the presented study is carefully conducted and of interest. It identifies *Cited4* as mediator of rosiglitazone-induced UCP1 expression in adipocytes.

Major comments

1) TZDs are mentioned to be potent insulin sensitizers. Such statement holds true for experiments in mice. However, experience in humans were quite disappointing. This needs to be emphasized accordingly.

2) Are findings reported specific for the PPAR γ ligand rosiglitazone or do other TZDs have a similar effect? At least some of the in vitro effects should be repeated using other TZDs/PPAR γ ligands. Such experiments would strengthen the reported findings since rosiglitazone is no longer a safe treatment option for type 2 diabetic subjects.

3) Were experiments in human SVF cells (Figs. 1D and E) performed in cells isolated from male and/or female subjects? Similar to experiments in SVF isolated from mice (Fig. 1C and Appendix Fig 1B), it would be important to perform experiment in SVF cells isolated from male and female human subjects in order to assess whether sex-specific difference is conserved between species.

4) The effect of *Cited4* knockdown on UCP1 expression was significant in female but not in male mice (Figs. 3C, 3D, 5A and 5B). However, UCP1 mRNA expression was also reduced by 50% in male mice. Such fact should be discussed in the revised manuscript. Moreover, authors should assess UCP1 protein levels in HFD-fed male and female mice, similar to experiments in chow-fed mice (Figs. 3C and D).

5) As outlined in Figure 2, Rosiglitazone-treated *Cited4* knockout cells revealed reduced expression of genes involved in oxidative phosphorylation. Was such pathway affected in *Cited4* KO mice treated with Rosiglitazone? It would be important to confirm this finding in vivo.

Minor comments

1) Rosiglitazone should not be written in capitals.

2) On several occasions e.g. in the abstract the term diabetes is used instead of the term type 2 diabetes (e. g. "current treatment options in diabetes"). Please replace accordingly since otherwise the statement made is not correct.

1st Revision - authors' response

27 April 2018

Referee #1 (Comments on Novelty/Model System for Author):

This concern is described in my comments to the authors

Referee #1 (Remarks for Author):

*In the present work, the authors describe the participation of the cofactor *Cited4* in the modulation of adipocyte differentiation from adipocyte progenitors upon rosiglitazone stimulation. The authors claim that this cofactor is involved in the rosiglitazone-mediated browning of scWAT. Surprisingly, the observed effects are not only tissue-specific (confined to scWAT), but also gender-restricted (occurring in females only). Unfortunately, the authors fall short in identifying the mechanisms behind the observed phenotype, making the work largely descriptive.*

*Moreover, many of the claims are not supported by the current data. The being phenotypes observed in vitro from adipocyte progenitors and in vivo from subcutaneous fat pads after rosiglitazone treatment are only based on gene expression analysis. Functional analyses should be performed to evaluate if changes found in gene expression translate into enhanced mitochondrial biogenesis, uncoupling, or fatty acid oxidation. Effects on energy expenditure are only investigated in living animals. This is a major concern given the ubiquitous expression of *Cited4* and the type of mouse model used (whole body KO). A similar flaw can be formulated for the insulin sensitization*

experiments presented in the manuscript. It cannot be concluded that the impaired insulin sensitization upon therapeutic rosiglitazone treatment in Cited4^{-/-} mice are mediated by white adipose tissue since Cited4 is expressed in all major metabolic or endocrine organs (brown adipose tissue, skeletal muscle, liver etc...). Animals with white adipose tissue specific deletion must be used to reinforce the authors' conclusions.

Finally, the observation that gender-specific effects of Cited4 contribute to the phenotype is interesting, yet the authors provide little discussion about the possible mechanisms underlying this effect. Altogether, this manuscript leaves many questions unexplained and unexplored.

We have performed cellular respiration analysis of Cited4 deficient adipocytes (new Fig 2), which revealed a defect specifically in uncoupled mitochondrial respiration in response to a beta3-adrenoreceptor agonist targeting adipocytes. This is consistent with the reduced Ucp1 mRNA and protein expression shown in cultures and in scWAT as well as with the reduction in energy expenditure upon beta3-adrenoreceptor stimulation in vivo.

New data show that both the energy expenditure and the insulin sensitivity phenotypes are only evident under rosiglitazone treatment. Given that extensive evidence suggests that adipose tissue is the major target of TZDs (reviewed by Soccio et al, 2014; Ahmadian et al, 2013), it is plausible to suggest that the associations between the scWAT defects (incl. ex vivo) and energy expenditure/insulin sensitization phenotypes are linked. We recognize though that we cannot provide definitive proof for this and have addressed this weakness in the Discussion (page 8). We would like to mention that to our knowledge there are currently no Cre mouse lines available which would enable Cited4 inactivation specifically in white adipose tissue including immature progenitor cells:

- Adipoq-Cre: Does not target progenitors; targets adipocytes in brown fat (in addition to scWAT and gWAT).
- Pparg-Cre: Targets non-adipogenic cells such as immune cells and endothelial cells, which can contribute to TZD-mediated effects.
- Pdgfra/Pdgfrb-Cre: Targets vascular and mesenchymal cells throughout the body incl. the brain.
- Acta2-Cre: Targets vascular cells throughout the body.
- Prx1-Cre: Targets skeletal muscle and bone, both metabolically relevant.

Finally, we have included a thorough discussion of the potential mechanisms for the involvement of Cited4 in sex-specific regulation (page 9).

Referee #2 (Comments on Novelty/Model System for Author):

Overall the work appears well done technically. The novelty of another transcriptional co-regulator in adipocytes is not high, especially given the lack of molecular mechanism. Medical impact of these finding is remote, as much more would need to be elucidated. The most interesting observations (i.e. effects of Cited4 in females not males, subcutaneous but not other fat depots, in response to TZD but not adrenergic stimulation) are purely descriptive.

We would like to thank the Referee for the detailed comments which have truly helped us to improve the manuscript.

Despite the weaknesses of our study we hope that the novelty is recognized, in the sense that the extensive current understanding of transcriptional networks in adipocytes does not comprehensively cover sex, tissue and stimulus-specific regulation. We have emphasized this in the Discussion (page 8).

Referee #2 (Remarks for Author):

Bayindir-Buchhalter et al submit a manuscript "Cited4 is a sex-based mediator of the antidiabetic glitazone response in adipocyte progenitors". They identified Cited4 by global profiling of mouse adipocyte precursors as a transcript induced by the PPAR γ agonist cPGI2, and showed similarly transient induction during adipogenesis stimulated by rosiglitazone. A Cited4^{-/-} mouse model was developed, and female adipocyte precursors from these mice show a selective defect in expression of beige adipocyte genes (though not general adipocyte markers). This was confirmed in human SVF-derived adipocytes with siRNA knockdown of CITED4, and in inducible Cre-mediated knockout of

Cited4 *ff/ff* mouse cells in culture. Expression profiling of wild type vs knockout adipose progenitors at 2 days of differentiation was consistent with their model that *Cited4* is required for maximal expression of genes in beige adipocyte pathways. Whole body knockout mice were also studied, and while there was no gross difference in adipogenesis, the beiging response of scWAT (particularly *Ucp1* mRNA and protein expression) was blunted in female knockout mice. Interestingly, this phenotype was not observed in male mice, nor in gonadal WAT or interscapular BAT, nor when beiging was induced by cold or beta agonists. Rosi-treated female *Cited4* knockout mice had a mild defect in energy expenditure despite similar physical activity and food intake, consistent with less uncoupled respiration. Finally, rosi-treated female knockout mice on high fat diet had impaired glucose and insulin tolerance, consistent with insulin resistance despite similar body weights. This was not observed in male knockouts, again indicating a sex-selective requirement for *Cited4* in rosi-mediated scWAT beiging.

Overall these are interesting observations and it is a well-written manuscript, but there are a number of issues to address:

1) Figure 1A-B nicely show transient induction of *Cited4* during adipogenesis from precursor cells isolated from mouse adipose tissue. However, given the sex differences described later in the paper, the sex is not clearly described in these figures. Was the *Cited4* induction sex-dependent? Also, the mouse 3T3-L1 cell line is very commonly used to study adipocyte differentiation, and it would be valuable to know whether *Cited4* is also transiently induced in a rosi-dependent manner in this model as well.

We have indicated the sex for all panels in Figure 1. We have included the following analyses:

- Time course analysis of progenitor cells from male mice (Fig EV1)
- Time course analysis of human primary SVF with indication of the sex of the patients (Fig 1).
- Time course analysis of 3T3-L1 as well as C3H10T1/2 cells (adipogenic differentiation) (Fig EV1).

The induction of *Cited4* by Rosi is not restricted to females. The expression pattern of primary mouse cells was recapitulated in the C3H10t1/2 line but not in 3T3-L1 cells, which showed transient induction by differentiation rather than by Rosi.

2) *Cited4* *ff/ff* and *-/-* mice are described for the first time. The targeting strategy is described somewhat in the methods, but this manuscript would benefit from a supplemental figure detailing the constructs, genotyping, and validation. For instance, I was confused how the entire ORF could be used in the targeting construct until I looked on a genome browser and saw *Cited4* is a small single exon gene.

We have prepared a figure with a schematic of the targeting strategy and validation by *Cited4* Western blots (Appendix Fig S1). Also, we have included genotyping information in Methods.

3) Figure S1A and S4A make it clear that *Cited4* mRNA is undetectable in SQ fat from *-/-* mice. For this initial report of the knockout, ideally this should also be demonstrated at the protein level. A brief internet search revealed commercial antibodies (i.e. Anti-CITED4 antibody, AbCam ab105797) that reportedly detect the protein in mouse tissues like heart. Notably, mouse GeneAtlas shows similar *Cited* transcript abundance in muscle as in white fat, so it would also be valuable to demonstrate loss of transcript and protein in this tissue. (Also, is there a cardiac phenotype to these whole body knockout mice?) It should also be noted in the discussion that the whole body knockout would also result in loss of expression in non-adipose tissues like skeletal muscle, which could be relevant to insulin sensitivity. Tissue-specific and inducible knockouts are beyond the scope of this paper but could be mentioned as future directions. Furthermore, *Cited4* is a member of gene family with *Cited1-3*, and no mention is made of these others members: their expression, regulation, potential functional compensation, etc.

Despite difficulties with *Cited4* antibodies over several years, we have achieved to detect the protein and to show complete loss of the protein upon *Cited4* knockout in scWAT progenitor cells, heart and scWAT tissue (Appendix Fig S1). By inference and given the targeting strategy (deletion of complete coding sequence), we assume that the knockout occurs in all tissues.

We have included data documenting the absence of gross differences between genotypes in basic cardiovascular parameters, i.e. heart weight, pulse and blood pressure, at least under steady state (Appendix Table S2).

We have addressed the lack of tissue specificity in our knockout model as a weakness of the study in the Discussion (page 8). We would kindly ask the Referee to also read the response to the corresponding comment of Referee 1.

We have included a discussion with literature references on the relevance of Cited1 and Cited2 (page 9).

4) Figure 1C looks at a panel of brown/beige genes in the cultured cells, yet omits Elov13 which in later figures is the gene most affected by Cited4 knockout in adipose tissue.

We have included Elov13 in Fig 1C (showing a trend of reduction).

5) Figure 1D shows a marked induction of Cited4 in rosi-treated human SVF-derived adipocytes. However, it is unclear whether this is an effect of rosi or simply adipocyte differentiation. Figure 1C shows that expression of the adipocyte marker adiponectin is ~100X less in the absence of rosi (note log scale), whereas in mouse precursors the difference was only ~5X (Figure 1C). This could reflect the fact that human adipocyte models generally require a TZD for full differentiation, making it challenging to study human cultured adipocytes in the absence of drug. Furthermore, in mouse adipocytes Cited4 expression was transient during adipogenesis and strikingly returned to basal levels in mature adipocytes (Figure 1A-B). Rather than a single day 9 time point, a time course of human adipogenesis would be necessary to show this pattern in human cells.

We have included a time course analysis of CITED4 expression in human adipogenesis over 14 days (Fig 1; Fig EV2). A trend of reduction of CITED4 expression between days 10 and 14 is visible, which may reflect the slower differentiation of human SVF cells compared to mouse cells.

We have included phase contrast images of the differentiated cultures showing that differentiation does occur to a considerable extent in the absence of Rosi (Fig EV2). In addition, we show that in the absence of Rosi, CITED4 expression correlates with UCP1 expression rather than ADIPOQ (Fig 1). Nevertheless, since Pparg activity is known to be increased during differentiation even in the absence of Rosi, we would expect a certain increase in Cited4 expression, which was actually observed in all cellular models. In any case, we have commented on the differences between the human and mouse cell models incl. the dependence on Pparg agonism for differentiation in Results and Discussion.

6) Figure 1E uses two different siRNAs to knock down expression of Cited4, yet the two have different effects on some genes (Adipoq and Slc2a4). The suppression of Adipoq by siCited4.1 is inconsistent with the author's model that Cited4 is specific for "beige" but not "white" adipocyte genes, so this deserves comment.

Based on the current data it is difficult to determine the reason for the discordance between siRNAs. Since PPARG mRNA was also slightly reduced by both siRNAs, we speculate that in human cells, at least in culture, CITED4 may also be essential for full general adipogenesis, possibly due to the higher dependency of human cells on Pparg agonists.

We have commented this point in Results and Discussion.

7) Figure 1I should be on two separate graphs rather than left and right axes, which are confusing.

We have rearranged the graph as suggested.

8) The expression profiling in figure 2 would benefit from more gene level data in addition to the pathway analyses presented. For instance, heat maps, Venn diagrams, and/or volcano plots of regulated genes would add to the analysis, particularly with identification of key genes like Ucp1, Cidea, Elov13, etc. Also, the focus is entirely on rosi-induced genes, and figures like these would show whether rosi-repressed genes are similarly affected by Cited4 knockout. Finally, the profiling

only at day 2 progenitors may be misleading, as it remains possible that the differences in gene expression do not persist in mature adipocytes.

We have presented an MA-plot which visualizes differential expression and expression levels and highlighted relevant genes (Fig 3).

We have prepared Venn diagrams for the overlapping regulation by Cited4 knockout and Rosi at the gene and pathway level incl. genes/pathways up-/down-regulated by Rosi, as suggested.

We have included qRT-PCR expression data from cells on day 8 of differentiation on genes shown in the MA-plot incl. some defining genes in the enriched OxPhos pathway (Fig 1). The expression patterns appear to be similar at day 2 and 8. We would like to note though that the main purpose of the profiling at day 2 was to identify Cited4-dependent gene sets incl. transcription factor target gene sets in order to uncover pathways regulated by Cited4 at its peak of Rosi-induced expression (as opposed to characterizing the metabolic phenotype of Cited4-deficient adipocytes).

9) In Figure 3B, in male scWAT it appears that, in the absence of rosi, loss of Cited4 clearly decreases expression of all 5 "beige" genes (comparing the first two bars). Was this effect significant? If so, it is also inconsistent with the model that Cited4 specifically affects only rosi-mediated gene expression.

Indeed, the expression of several of the beige genes in male mice without Rosi were significantly different between genotypes (please note n=4 in one male vehicle group). We have focused our investigation throughout the manuscript on Rosi-treated cells/mice incl. statistical testing (as mentioned in the Methods section) and report the control (no-Rosi) groups mainly descriptively, at least for expression data. "Browning" in BL6 mice at room temperature is very low compared to stimulated conditions (Rosi, cold etc). We could not detect Ucp1 protein in scWAT of Control (no-Rosi) males. This also applies to progenitor differentiation in the absence of Rosi. In addition, we have not followed up the indicated difference because we did not detect genotype differences in insulin sensitivity of males with or without Rosi or in male progenitor cells. Of note, CL treatment did not reveal a genotype difference in scWAT expression in males either (data not shown). Thus, we would like to clarify that our claim on the Rosi-specific function of Cited4 refers to the comparison to other conditions of considerable browning, i.e. cold and beta3-adrenergic (CL) stimulation.

10) In Figure 3A-B, the effects of Cited4 knockout on Ucp1 mRNA expression in scWAT are a similar 2-2.5x in both females and males, yet the in 3C the protein effects are markedly different. The reasons for this disconnect between RNA and protein are not discussed. Also, the Western Blots in figure 1C differ between the sexes, with a much fainter band in females along with the appearance of an apparent nonselective band below Ucp1. Given this result, it would be valuable to directly compare male and female scWAT on the same blot to investigate the sex difference.

We have included a graph to show that the correlation between Ucp1 mRNA and protein appears to be different between females and males (Appendix Fig S4 and comment on page 6). Whether this is generally applicable and reflects differential regulation of Ucp1 protein expression remains to be determined.

The non-specific band appears variably in the Ucp1 blots in our hands. We have resolved the specificity of the Ucp1 bands by including samples from Ucp1 knockout and wild type BAT in a separate blot (Appendix Fig S5).

We have included a blot with female and male samples showing slightly increased Ucp1 expression in males compared to females under Rosi but have not further explored this due to time constraints (Fig EV3).

11) Figure S4 in the appendix contains key data that belongs in the main manuscript. Figure S4A shows that Cited4 mRNA is not rosi-induced in scWAT, which deserves mention given the effect of rosi on cultured cells in Figure 1. The expression data in S4B-C is also very relevant but needs some additional pieces: all three fat depots need to be compared in both sexes, as well as cardiac/skeletal muscle where the RNA is also abundant. Figures S4D and S4E show the key and very interesting

negative results (even mentioned in the abstract) that Cited4 does not affect scWAT beiging by beta3-agonist or cold exposure. Other negative data in main Figure 3 (i.e. 3E and 3F showing no effect of knockout in other depots) could be moved to supplemental to make room for this more relevant data in the main text. Also, figures S4D and S4F would benefit from included the unexposed scWAT as a control, to confirm the extent of beiging by drug and cold (similar to the absence of rosi in the rest of figure 3).

The Cited4 expression data in scWAT +/-Rosi (former Fig S4A) have been moved to new Fig 4A and commented on page 6.

We have included a graph comparing Cited4 expression in scWAT, gWAT, BAT, muscle, heart and liver of females and males (new Fig EV3).

We have included non-exposed wild type control groups in the analyses of scWAT under cold and beta3-agonist exposures (former Fig S4D and E) and have moved the graphs to new Fig 4.

We have moved gWAT/BAT data (former Fig 3E and F) to new Fig EV3.

12) Figure 4 shows a significant difference in oxygen consumption in female mice in the presence of rosi, as this is the sex and treatment proposed to be relevant by the authors' model. However, it would be valuable to know whether this difference exists the absence of rosi, or in male mice.

We have included oxygen consumption data on (i) female wild type/knockout mice in the absence of Rosi (new Fig EV3) and (ii) male mice in the presence of Rosi incl. acute beta3-adrenoreceptor agonist (CL) stimulation (new Fig 5). To ensure that our indirect calorimetry data comply with the highest current standard we have applied ANCOVA analysis for adjustment of VO₂ values to body weight instead of normalization to body weight (Tschöp et al, 2011). In this sense, we have also replaced the previous graph on females+Rosi (former Fig 4B) with ANCOVA-adjusted data (new Fig 5B).

The results show that the reduction of energy expenditure by Cited4 knockout is restricted to female mice under Rosi (+/- CL) and strengthen the proposed association between compromised induction of Ucp1 by Rosi in scWAT of Cited4-knockout mice with defects in systemic energy expenditure and metabolism.

13) Similarly, Figure 5 shows HFD-fed mice only after treatment with rosi, but does not show the effect of rosi treatment. This is particularly relevant for the ITT in Fig 5D, in which there is minimal response to insulin in the control female mice and virtually none in the Cited4 knockouts. This indicates the mice were extremely insulin resistant yet rosi should have been insulin sensitizing. Including an untreated but HFD-exposed group in this experiment would be necessary to show the insulin sensitizing effect of rosi, and to make a more convincing case that the effect is lost in the Cited4 knockout mice. Finally, given how important the insulin resistance of the knockout mice is to the overall conclusion of the manuscript, the gold standard assay of insulin sensitivity (hyperinsulinemic-euglycemic clamp) would be a much better test than an ITT.

We have included new data from an independent experiment to address this point, in which we have used a higher dose of insulin resulting in more informative ITT data (new Fig 7). The compromised response to insulin in Rosi-treated Cited4-knockout females (but not males) was reproduced and was not evident in the same HFD-fed mice before the start of Rosi treatment. Furthermore, we observed that the improvements caused by Rosi treatment in the ITT response, fasting insulin and interestingly serum fatty acids in wild type mice were not significant in Cited4 knockout mice. These genotype differences were specific to females.

We recognize that clamp experiments would have been more informative but have not been able to perform these in the given time frame due to logistical reasons related to approval of new mouse procedures by the authorities, coordination with partners for setting up the assay and availability of knockout mice in sufficient numbers. We feel though that the new independent evidence on insulin sensitivity has consolidated our conclusions substantially.

14) Finally, no mechanism is explored or even proposed for *Cited4*. The gene name "*Cbp/p300* interacting transactivator with *Glu/Asp* rich carboxy-terminal domain 4" indicates that it is itself a transcriptional co-regulator, so could its effects on gene regulation be direct as part of transcriptional regulatory complex with PPAR γ ? There is also literature (briefly mentioned in the introduction) about *Cited4* and C/EBP transcription factors in cardiac hypertrophy, yet C/EBPs are also highly relevant in adipogenic gene regulation. Even if exploration of the molecular mechanism of *Cited4* is beyond the scope of this paper, and brief review of *Cited4* known biology and its potential role in adipocytes should be discussed.

Reporter assays with *Cited4* overexpression do not indicate that *Cited4* is a simple direct co-activator of Pparg at least not in the context of the isolated PPRE response element. This would be in agreement with the selective regulation of Pparg target genes by *Cited4* knockout. However, these results are preliminary and have not been included in the manuscript. We have included a section in the Discussion (page 9) in which we speculate about possible mechanisms of sex-specific regulation by *Cited4* and the functional interaction with Pparg based on known facts about *Cited4* and its family members. We have not included *Cebpb* in this discussion as the mentioned studies imply that *Cebpb* is upstream of *Cited4*, an aspect which we haven't addressed.

Referee #3 (Remarks for Author):

Bayindir-Buchhalter et al. investigated the role of the PPAR γ transcriptional cofactor Cited4 as a target and mediator of rosiglitazone in adipocyte progenitor cells. They report that Cited4 is required for rosiglitazone-mediated induction of thermogenic expression in subcutaneous fat mainly in female mice. Importantly, Cited4 appears not to be involved in beta-adrenergically or cold stimulated browning.

Overall, the presented study is carefully conducted and of interest. It identifies Cited4 as mediator of rosiglitazone-induced UCP1 expression in adipocytes.

Major comments

1) TZDs are mentioned to be potent insulin sensitizers. Such statement holds true for experiments in mice. However, experience in humans were quite disappointing. This needs to be emphasized accordingly.

We'd like to thank the Referee for the insightful and constructive comments. To our knowledge, there is solid evidence for the efficacy of TZDs in treating type 2 diabetes and for insulin sensitization being a major mechanism of action in humans (for instance, Yau H, *Curr Diab Rep*, 2013; Natali A, *Diabetologia*, 2006; Kahn SE, *N Engl J Med*, 2006; DeFronzo RA, *N Engl J Med*, 2011). In our understanding, the failure of the TZDs in the clinics is due to their side effects. To clarify this we have now stated in the introduction "...the use of TZDs has *strongly* declined due to their side effects". The reviews we have cited provide an overview of the risks and benefits of TZDs. However, we would welcome suggestions for major studies disproving the efficacy/insulin sensitization by TZDs in humans which we could cite in our manuscript.

2) Are findings reported specific for the PPAR γ ligand rosiglitazone or do other TZDs have a similar effect? At least some of the in vitro effects should be repeated using other TZDs/PPAR γ ligands. Such experiments would strengthen the reported findings since rosiglitazone is no longer a safe treatment option for type 2 diabetic subjects.

We have included data on the *Cited4* knockout cell phenotype under pioglitazone treatment, representing the main TZD currently used for type 2 diabetes treatment. We observed an induction of *Cited4* by pioglitazone during differentiation of adipocyte progenitors, in accordance with the effect of rosiglitazone (new Fig EV1). Furthermore, we examined the effects of *Cited4* knockout on thermogenic gene expression during progenitor differentiation under pioglitazone in direct comparison to rosiglitazone and observed a comparable phenotype (new Fig EV1). Of note, pioglitazone effects on *Cited4* and thermogenic expression occurred with an approximately 10-fold lower potency compared to rosiglitazone, in accordance with the known reduced Pparg activation potency.

3) *Were experiments in human SVF cells (Figs. 1D and E) performed in cells isolated from male and/or female subjects? Similar to experiments in SVF isolated from mice (Fig. 1C and Appendix Fig 1B), it would be important to perform experiment in SVF cells isolated from male and female human subjects in order to assess whether sex-specific difference is conserved between species.*

We have included new data showing the induction of CITED4 expression by rosiglitazone in both female and male cells, similarly to the mouse (new Fig 1). The knockdown data shown in the previous manuscript version were from female cells. We have included data from male cells in the revised version, which show that the effects of CITED4 knockdown were evident independent of sex (new Fig 1). Whether the differences compared to the mouse system were due to the use of siRNAs, the depot location or the stronger dependence of human SVFs on Pparg agonists for differentiation remains to be determined. The phenotypic differences between species are discussed on page 9.

4) *The effect of Cited4 knockdown on UCP1 expression was significant in female but not in male mice (Figs. 3 C, 3D, 5A and 5B). However, UCP1 mRNA expression was also reduced by 50% in male mice. Such fact should be discussed in the revised manuscript. Moreover, authors should assess UCP1 protein levels in HFD-fed male and female mice, similar to experiments in chow-fed mice (Figs. 3C and D).*

We have included a graph to show that the correlation between Ucp1 mRNA and protein appears to be different between females and males (Appendix Fig S4 and comment on page 6). Whether this is generally applicable and reflects differential regulation of Ucp1 protein expression remains to be determined.

We have included Ucp1 Western blot analysis of scWAT from the HFD/Rosi-fed mice (new Fig 6). The experiment was influenced by few strong outliers (plotted in the graph) but a trend of Ucp1 reduction was detected in female but not male Cited4-knockout mice.

5) *As outlined in Figure 2, Rosiglitazone-treated Cited4 knockout cells revealed reduced expression of genes involved in oxidative phosphorylation. Was such pathway affected in Cited4 KO mice treated with Rosiglitazone? It would be important to confirm this finding in vivo.*

We have selected several genes from the OxPhos gene set (Cyc1, Cox8b, Cox7a, Ndufb3) and have highlighted their differential expression in a new MA-plot from the expression profiles from cells (new Fig 3). We have measured their expression by qRT-PCR in scWAT and observed a sex-specific reduction by Cited4 knockout in several cases in both in vivo models (new Fig 4 and Fig 6).

Minor comments

1) *Rosiglitazone should not be written in capitals.*

2) *On several occasions e.g. in the abstract the term diabetes is used instead of the term type 2 diabetes (e. g. "current treatment options in diabetes"). Please replace accordingly since otherwise the statement made is not correct.*

We have corrected the text according to these comments.

Thank you for the submission of your revised manuscript to EMBO Molecular Medicine. We have now received the enclosed reports from the referees that were asked to re-assess it. As you will see the reviewers are now globally supportive and I am happy to inform you that we will be able to accept your manuscript pending the following final amendments:

1) Please address the comments of referee 1 in writing, acknowledging the limitations of the study in the discussion section. At this stage, we'd like you to discuss referee's 1 points and if you do have data at hand, we'd be happy for you to include it, however we will not ask you to provide any additional experiments.

Please provide a letter INCLUDING my comments and the reviewer's reports and your detailed responses to their comments (as Word file).

Please submit your revised manuscript within two weeks. I look forward to seeing a revised form of your manuscript as soon as possible.

***** Reviewer's comments *****

Referee #1 (Comments on Novelty/Model System for Author):

See my comments first submission and revised version

Referee #1 (Remarks for Author):

Altogether, the authors have improved their manuscript but the molecular mechanism remains elusive. One major drawback remains the fact that the contribution of adipose tissue *Cited4* to the observed phenotype is only suggested by the experimental strategy presented here. We agree with the authors for the lack of a Cre mouse line that would enable *Cited4* inactivation specifically in white adipose tissue immature progenitor cells. However, using an *Adipoq*-Cre line would have helped in narrowing the phenotype down to adipose fat pads (likely white) and would have taken the potential role of the muscle out of the equation. If no phenotype would have been observed in *Cited4**Adipo*^{-/-}, it would have at least suggested that the effects of *Cited4* on the browning program take place before differentiation of the cells. On the other hand, it needs to be stressed that the expression of *Cited4* in the muscle is really high and several papers reported an effect of rosiglitazone in the improvement of insulin sensitivity, fatty acid oxidation and uncoupling in the skeletal muscle (Schrauwen P, JCEM, 2006; Liu Y, Am J Physiol Endocrinol Metab, 2009; Kim JK, Diabetes, 2003, Mensink M, Int J Obesity, 2007). If we consider that the mass of skeletal muscle compared to scWAT is much higher, it is possible that the effects on insulin sensitivity and energy expenditure are mediated by this tissue. The studies with the full KO cannot discard this possibility. Furthermore, some seamlessly important results are left on the side such as the downregulation of browning genes in scWAT of *Cited4*^{-/-} males only in the absence of Rosi. These non-studied results suggest that the role of *Cited4* in controlling the thermogenic program of white adipose tissue might not be as straightforward as one could imagine. Finally, the authors also decided not to dig into the gender specific effects of *Cited4*, which is clearly the main finding of this study. Although the authors suggest a possible role of estrogen receptor signaling, at this point associations are not enough to claim prove of concept.

Referee #2 (Remarks for Author):

The authors did a nice job addressing the concerns raised in my initial review. Their discussion now acknowledges the lack of mechanism but focuses on the novelty and potential relevance of the descriptive findings. They also address the limitations of whole body knockout but make a case for adipose-specific effects. New data, particularly the Seahorse assay (Fig 2F-G) and ITT (Fig 7E-F), strengthen the conclusions. The expression profiling data analysis in Figure 3 is also much improved. The Western blots, cardiac phenotyping, and other data in the appendix are also valuable additions. Appropriate clarifications regarding sex of cell donors, etc were made and generally support the author's model. This manuscript appears suitable for publication with minor revisions.

A few minor issues:

-in Figure 1E-F the correlation or lack thereof is apparent to the eye, but a statistical test (i.e. R-squared value) would support the conclusion.

-in Figure 7B, the authors focus on the fact that rosi doesn't lower fasting insulin in the knockout females. However, this incorrectly implies that the insulin level stays high with drug treatment in the diet-induced obese mice. What the data actually show is lower insulin in the absence of drug, i.e. less diet-induced insulin resistance rather than less drug response. This is a different result than the ITT in Figure 7E.

-in the discussion, the authors suggest estrogen receptor as a potential mediator of the sex-selective effects. With this in mind, it bears mentioning that the inguinal fat pad (the subcutaneous depot that shows all the differences in this manuscript) develops mammary glands in female mice.

Referee #3 (Remarks for Author):

The authors did a very good Job to address my concerns.

2nd Revision - authors' response

07 June 2018

***** Reviewer's comments *****

Referee #1 (Comments on Novelty/Model System for Author):

See my comments first submission and revised version

Referee #1 (Remarks for Author):

Altogether, the authors have improved their manuscript but the molecular mechanism remains elusive. One major drawback remains the fact that the contribution of adipose tissue Cited4 to the observed phenotype is only suggested by the experimental strategy presented here. We agree with the authors for the lack of a Cre mouse line that would enable Cited4 inactivation specifically in white adipose tissue immature progenitor cells. However, using an Adipoq-Cre line would have helped in narrowing the phenotype down to adipose fat pads (likely white) and would have taken the potential role of the muscle out of the equation. If no phenotype would have been observed in Cited4Adipo^{-/-}, it would have at least suggested that the effects of Cited4 on the browning program take place before differentiation of the cells. On the other hand, it needs to be stressed that the expression of Cited4 in the muscle is really high and several papers reported an effect of rosiglitazone in the improvement of insulin sensitivity, fatty acid oxidation and uncoupling in the skeletal muscle (Schrauwen P, JCEM, 2006; Liu Y, Am J Physiol Endocrinol Metab, 2009; Kim JK, Diabetes, 2003, Mensink M, Int J Obesity, 2007). If we consider that the mass of skeletal muscle compared to scWAT is much higher, it is possible that the effects on insulin sensitivity and energy expenditure are mediated by this tissue. The studies with the full KO cannot discard this possibility. Furthermore, some seamlessly important results are left on the side such as the downregulation of browning genes in scWAT of Cited4^{-/-} males only in the absence of Rosi. These non-studied results suggest that the role of Cited4 in controlling the thermogenic program of white adipose tissue might not be as straightforward as one could imagine. Finally, the authors also decided not to dig into the gender specific effects of Cited 4, which is clearly the main finding of this study. Although the authors suggest a possible role of estrogen receptor signaling, at this point associations are not enough to claim prove of concept.

We are grateful to Referee 1 for the constructive discussion. Regarding the tissue-specific functions of Cited4, we have not proceeded with the examination of the adipocyte-specific knockout phenotype (Adipoq-Cre) since we did not observe any considerable effect on thermogenic/adipogenic gene expression upon Cre-mediated Cited4 inactivation in committed differentiating progenitors/adipocytes in cell culture (Fig EV2G), which is consistent with the downregulation of Cited4 expression during adipocyte differentiation (Fig 1B). There is no doubt about the need for adipocyte progenitor-specific knockout experiments and we have clearly acknowledged this in the Discussion. We have included the suggested citations on the response of skeletal muscle to TZDs in the discussion, emphasizing the potential role of Cited4 function in muscle (page 8). However, we have also clarified that the two reports using myocyte-specific Pparg deletion to address the role of muscle in systemic insulin sensitization by TZDs are contradictory.

Regarding the downregulation of browning genes in scWAT of Cited4^{-/-} males in the absence of Rosi, we have included a comment in the Discussion (page 9). As pointed out in our response to the corresponding comment of Referee 2, the relevance of white adipose tissue thermogenesis under normal conditions (room temperature) in BL6 mice is probably questionable. We could not detect Ucp1 protein in scWAT in the absence of Rosi and we could not find associations between

differences in thermogenic expression and insulin sensitivity in males with or without Rosi (or energy expenditure under Rosi treatment). In addition, we did not observe genotype differences in CL-stimulated thermogenic expression in scWAT (data not shown but available upon request).

We find that while several important questions regarding the function of Cited4 have emerged, our study presents very consistent associations between cellular phenotypes, scWAT phenotypes, energy expenditure and insulin sensitization, revealing novel aspects of sex-dependent regulation of metabolism in the therapeutically relevant context of Pparg targeting.

Referee #2 (Remarks for Author):

The authors did a nice job addressing the concerns raised in my initial review. Their discussion now acknowledges the lack of mechanism but focuses on the novelty and potential relevance of the descriptive findings. They also address the limitations of whole body knockout but make a case for adipose-specific effects. New data, particularly the Seahorse assay (Fig 2F-G) and ITT (Fig 7E-F), strengthen the conclusions. The expression profiling data analysis in Figure 3 is also much improved. The Western blots, cardiac phenotyping, and other data in the appendix are also valuable additions. Appropriate clarifications regarding sex of cell donors, etc were made and generally support the author's model. This manuscript appears suitable for publication with minor revisions.

We thank Referee 2 for the substantial support and concrete comments to improve our manuscript.

A few minor issues:

-in Figure 1E-F the correlation or lack thereof is apparent to the eye, but a statistical test (i.e. R-squared value) would support the conclusion.

We have included the Pearson correlation coefficient and the corresponding p-value within the graphs and have amended the Results text to indicate the trend (P=0.08) for a strong positive correlation ($r=0.8$) between CITED4 and UCP1 mRNA.

-in Figure 7B, the authors focus on the fact that rosi doesn't lower fasting insulin in the knockout females. However, this incorrectly implies that the insulin level stays high with drug treatment in the diet-induced obese mice. What the data actually show is lower insulin in the absence of drug, i.e. less diet-induced insulin resistance rather than less drug response. This is a different result than the ITT in Figure 7E.

The difference between genotypes in serum insulin before Rosi is far from significant (P=0.725 in post test). We have now included this important info in the figure legend. Fig 7B and 7E represent independent experiments and we assume that the indicated effect before Rosi represents a cohort difference. The ITT after Rosi corresponding to the cohort in 7B was significantly different (Fig EV4C). We don't have ITT data before Rosi for this cohort or insulin data for the cohort of 7E.

-in the discussion, the authors suggest estrogen receptor as a potential mediator of the sex-selective effects. With this in mind, it bears mentioning that the inguinal fat pad (the subcutaneous depot that shows all the differences in this manuscript) develops mammary glands in female mice.

We have included this info in a comment in the Discussion.

Referee #3 (Remarks for Author):

The authors did a very good Job to address my concerns.

We'd like to thank Referee 3 for his efforts in reviewing our manuscript.

Corresponding Author Name: Alexandros Vegiopoulos

Journal Submitted to: EMBO Mol Med

Manuscript Number: EMM-2017-08613